



# Variability of OH reactivity in the Landes maritime Pine forest: Results from the LANDEX campaign 2017

Sandy Bsaibes[1], Mohamad Al Ajami[2], Kenneth Mermet[3,4,5], François Truong[1], Sébastien Batut[2], Christophe Hecquet[2], Sébastien Dusanter[3], Thierry Léornadis[3], Stéphane Sauvage[3], Julien Kammer[1,4], Pierre-Marie Flaud[4,5], Emilie Perraudin[4,5], Eric Villenave[4,5], Nadine Locoge[3], Valérie Gros[1], Coralie Schoemaecker[2]

[1] Laboratoire des Sciences du Climat et de l'Environnement, LSCE, UMR CNRS-CEA-UVSQ, 91191 Gif-sur-Yvette, France
[2] Laboratoire PhysicoChimie des Processus de Combustion et de l'Atmosphère, PC2A, UMR 8522, 59655 Villeneuve d'Ascq, France
[3] IMT Lille Douai, Univ. Lille – SAGE - Département Sciences de l'Atmosphère et Génie de l'Environnement, F-59000 Lille, France
[4] Univ. Bordeaux, EPOC, UMR 5805, F-33405 Talence Cedex, France
[5] CNRS, EPOC, UMR 5805, F-33405 Talence Cedex, France

*Correspondence to*: Sandy Bsaibes (sandy.bsaibes@gmail.com)

Valérie Gros (valerie.gros@lsce.ipsl.fr)

**Abstract.**

Total OH reactivity measurements were conducted during the LANDEX intensive field campaign in a coniferous temperate forest located in the Landes area, south-western France, during July 2017. In order to investigate inter-canopy and intra-canopy variability, measurements were performed inside (6m) and above the canopy level (12m), as well as at two different locations within the canopy, using a Comparative Reactivity Method (CRM) and a Laser Photolysis-Laser Induced Fluorescence (LP-LIF) instrument. The two techniques were intercompared at the end of the campaign by performing measurements at the same location. Volatile organic compounds were also monitored at both levels with a proton transfer-time of flight mass spectrometer and online gas Chromatography instruments to evaluate their contribution to total OH reactivity, with monoterpenes being the main reactive species emitted in this *Pinus pinaster* Aiton dominated forest. Total OH reactivity varied diurnally, following the trend of BVOCs of which emissions and concentrations were dependent on meteorological parameters. Highest levels of total OH reactivity were observed during nights with a low turbulence ($u^* \leq 0.2$ m/s) leading to lower mixing of emitted species within the canopy and thus an important vertical stratification, characterized by a strong concentration gradient. By comparing the calculated OH reactivity from contributions of individually measured compounds to the measured OH reactivity, a discrepancy was seen at both heights mainly related to ambient temperature during day-time. It highlights that missing OH sinks could be due to temperature-dependent missing primary emissions or secondary products linked to a temperature-enhanced photochemistry. During night-time hours, atmospheric stability and relative humidity played a key role in the missing reactivity. Lower turbulence showed to be favourable for night-time chemistry, inducing a higher missing OH



reactivity. Humid surfaces may have represented an additional sink for oxygenated compounds, escaping de facto total OH reactivity, and leading to lower or no missing OH reactivity.

## 1 Introduction

The hydroxyl radical OH is considered as the most important initiator of photochemical processes in the troposphere during day-time, and the prevailing "detergent" from local to global scales. It controls the lifetime of most trace gases and contributes to the self-cleansing power or so-called "oxidation capacity" of the atmosphere.

Even though the main primary source for OH in the lower troposphere is the photolysis of ozone at short wavelengths, the OH production and loss processes are numerous and difficult to quantify. Such losses involve several hundreds of chemical species and as many reactions to consider. In this respect, a direct measurement of total OH reactivity ($R_{OH}$) is of great interest to better understand the OH chemistry in the atmosphere and to investigate the budget of OH sinks in a particular environment. $R_{OH}$ is defined as the pseudo first-order loss rate (in $s^{-1}$) of OH radicals, equivalent to the inverse of the OH lifetime. It is the sum of the reaction frequencies of all chemical species reacting with OH, as shown in Eq. 1:

$$R_{OH} = \sum_{i=1}^{n} k_{OH+Xi} \cdot [Xi] \qquad \text{(Eq. 1)}$$

In this equation, a chemical reaction frequency for a species $X_i$ with OH ($R_{OH+Xi}$) is the product of its rate-coefficient $k_{OH}$ with its concentration $[X_i]$. The measured total OH reactivity can be compared with calculated values based on the sum of reaction frequencies as shown in Eq. 1 and for which the concentration of $X_i$ has been measured at the same location. Any significant discrepancy between measured and calculated OH reactivity explicitly demonstrates missing OH sinks, commonly called missing OH reactivity, and points out that potentially important unmeasured reactive species and chemical processes associated with these species may affect our understanding of OH atmospheric chemistry.

Two approaches have been used to measure the total OH reactivity. The first approach derives OH reactivity from direct measurements of OH decay rates due to its reaction with trace species present in ambient air introduced in a reaction tube. OH can be generated and detected differently according to 3 types of techniques: The Flow Tube-Laser Induced Fluorescence (FT-LIF, (Hansen et al., 2014; Ingham et al., 2009; Kovacs and Brune, 2001)), the Laser Photolysis-Laser Induced Fluorescence (LP-LIF, (Sadanaga *et al.*, 2004; Parker *et al.*, 2011; Amédro, 2012; Stone et al., 2016; Fuchs *et al.*, 2017)) and the Flow Tube-Chemical Ionization Mass Spectrometry (FT-CIMS, (Muller et al., 2018)). The second approach is called the Comparative Reactivity Method (CRM) and it consists in an indirect quantification of OH losses from the concentration change of a reference molecule that competes with ambient reactive species to react with artificially produced OH. The reference substance, pyrrole, is measured with a Proton Transfer Reaction Mass Spectrometer (PTR-MS, (Sinha *et al.*, 2008; Dolgorouky *et al.*, 2012; Michoud *et al.*, 2015)) or with a Gas Chromatograph-Photo Ionization Detector (GC-PID, (Nölscher et al., 2012)) or chemical ionization mass spectrometry (CIMS, (Sanchez et al., 2018)).

Both LP-LIF and CRM techniques were deployed in a Pine forest for this study, the instruments deployed are presented in more details below and a general description is provided here. In the LP-LIF method, OH is generated by laser



pulsed photolysis of ozone in a reaction tube, at typically 266 nm, followed by the rapid reaction of O($^1$D) with ambient water vapor. OH radicals react with ambient reactive species in the reaction tube and the concentration of OH decreases after the laser pulse. The air from the reaction tube is continuously pumped into a low-pressure detection cell where the OH decay is monitored by laser-induced fluorescence at a high time resolution (range of hundreds of μs) (Sadanaga et al., 2004). Compared

to flow-tube set-ups, lower flow rates of ambient air are needed in the LP-LIF technique (less than 10 L min$^{-1}$ compared to several tens of L min$^{-1}$). In addition, the use of O$_3$ laser photolysis instead of continuous water photolysis by lamps at 185 nm for OH generation, the latter being commonly used in FT-LIF or CRM, limits the spurious formation of OH from the reaction of HO$_2$ with ambient NO. However, in order to quantify wall loss reactions, an instrument zero has to be subtracted from all measurements, and a correction may have to be applied for the recycling of OH radicals in the presence of high NO levels

(Stone et al., 2016; Fuchs *et al.*, 2017).

In the Comparative Reactivity Method (CRM), ambient air, wet nitrogen and pyrrole are introduced into a glass reactor where OH radicals are produced by the photolysis of water vapor. The mathematical expression used to determine the OH reactivity of the analyzed sample is derived in terms of the initial concentration of pyrrole (C1), the background concentration of pyrrole reacting alone with OH (C2) and the concentration of pyrrole after competition with air reactants

(C3). The CRM exhibits several advantages compared to direct measurements techniques, like the commercial availability of PTR-MS and the need of a smaller sampling flow rate of ambient air (few hundreds of mL min$^{-1}$), which broadens the application of the technique to branch and plant enclosure studies. On the other hand, this indirect method requires a raw data processing with careful corrections for measurement artefacts related to humidity changes and secondary chemistry that can impact the pyrrole concentration (Sinha *et al.*, 2008; Michoud *et al.*, 2015).

A few inter-comparisons were reported in the literature for urban and remote areas (Hansen *et al.*, 2015; Zannoni et al., 2015; Sanchez *et al.*, 2018) and chamber experiments (Fuchs *et al.*, 2017) aiming at reproducing ambient conditions observed in various environments. The latter, including a large number of OH reactivity instruments (FT-LIF, LP-LIF, CRM) and conducted in the SAPHIR atmospheric simulation chamber, allowed to compare the performances of each technique. Results showed that OH reactivity can be accurately measured for a wide range of atmospherically relevant chemical conditions

by all instruments. However, CRM instruments exhibited larger discrepancies to calculated OH reactivity compared to instruments directly probing OH radicals, and these differences were more important in the presence of terpenes and oxygenated organic compounds.

Over the past two decades, OH reactivity measurements were conducted in various environments at the ground level using the available techniques: urban and suburban areas, forest areas, marine areas (Yang *et al.*, 2016; Dusanter and Stevens,

2017). A few aircraft measurements have also been carried out to complete ground-based observations (Brune et al., 2010). Many studies highlighted the interest of investigating OH reactivity in forest areas exhibiting large concentrations of biogenic VOCs (BVOCs) since BVOC emissions exceed anthropogenic VOCs by a factor of 10 at the global scale (Guenther et al., 1995). Results showed that our understanding of OH sinks in these environments was incomplete with observations of large missing OH reactivity ranging between 25% and 80%. Total OH reactivity appeared to be impacted by several factors such as





the forest type and the dominant emitted species, the seasonality, the canopy level as well as specific atmospheric conditions (Hansen et al., 2014; Nölscher et al., 2013; Praplan et al., 2019; Sanchez et al., 2018; Zannoni et al., 2017).

Among these biogenic hydrocarbons, monoterpenes represent a large class of $C_{10}H_{16}$ compounds, which are mainly emitted by conifers as well as broad-leaves trees. They can be oxidized by OH, ozone and the nitrate radical, leading to

atmospheric lifetimes ranging between minutes and days (Atkinson and Arey, 2003). The oxidation of primary BVOCs can therefore contribute to the formation of tropospheric ozone and secondary organic aerosols from the local to the regional scales, with oxidation products of BVOCs having a potential impact at a larger scale. Regarding coniferous forests, an averaged OH reactivity of 6.7 $s^{-1}$ was observed over a temperate Pine forest located in the southern part of the Rocky Mountains in the USA during summer 2008 (Nakashima et al., 2013). Measured OH reactivity exhibited a diurnal variation with minima during day-

time when MBO (2-methyl-3-buten-2-ol) was the main contributor, and maxima during night-time when the OH reactivity was dominated by monoterpenes. Approximately 30% of the measured OH reactivity remained unexplained and could be related to unmeasured or unknown oxidation products of primary emitted biogenic compounds. Another campaign also carried out in a temperate coniferous forest, located in the Wakayama Forest Research Station in Japan during summer 2014 (Ramasamy et al., 2016), showed comparable results with an average total OH reactivity of 7.1 $s^{-1}$. OH reactivity varied

diurnally with temperature and light, reaching a maximum at noon-time. Monoterpenes were the main drivers of the total OH reactivity in the considered ecosystem, accounting for 23.7%, followed by isoprene (17.0%) and acetaldehyde (14.5%). The missing OH reactivity (29.5 % on average) was found to be linked to light and temperature dependent unmeasured primary and secondary species.

In the present study, we report on the measurement of total OH reactivity from a field experiment conducted in the

Landes temperate forest, southwestern France. This work was part of the LANDEX project (LANDEX, i.e. the Landes Experiment: Formation and fate of secondary organic aerosols generated in the Landes forest) that aimed at characterizing secondary organic aerosol formation observed in this monoterpene-rich environment. The dominant tree species at the site is maritime pine, *Pinus pinaster* Aiton, which is known to be a strong emitter of α and β-pinene, leading to a diurnal concentration profile of monoterpenes characterized by maximum values at night and minimum values during daytime (Simon et al., 1994).

Nocturnal new particle formation episodes (NPFs) were reported in this ecosystem, suggesting the contribution of BVOC oxidation to the nucleation and growth stages of particles (Kammer et al., 2018).

Measurements of OH reactivity and trace gases were performed at two heights to cover the inside and above canopy, and at two different locations inside the canopy to investigate the intra-canopy variability. Two different instruments were deployed: the CRM from LSCE (Laboratoire des Sciences du Climat et de l'Environnement) that measured inside and above the canopy

and the LP-LIF from PC2A (Physicochimie des Processus de Combustion et de l'Atmosphère) that performed measurements inside the canopy. The deployment of two different instruments was a good opportunity to (i) compare measurements made with both methods in a real biogenic environment after the inter-comparison experiment performed in the SAPHIR chamber and recent improvement of the CRM instrument, (ii) investigate the levels and diurnal variability of OH reactivity at two



different heights, and (iii) investigate both the OH reactivity budget and the missing reactivity pattern using a large panel of concomitant trace gas measurements.

## 2 Experimental

### 2.1 Site description

The LANDEX intensive field campaign was conducted from the 3$^{rd}$ to the 19$^{th}$ of July 2017 at the Bilos field site in the Landes forest, south-western France. The vegetation on the site was dominated by maritime pines (*Pinus pinaster* Aiton) presenting an average height of 10 m. The climate is temperate with a maritime influence due to the proximity of the Atlantic Ocean. This site is part of the European ICOS (Integrated Carbon Observation System) Ecosystem infrastructure. A more detailed description of the site is available in Moreaux *et al.* (2011) and Kammer *et al.* (2018).

**2.2 OH reactivity instruments**

The LP-LIF instrument, referred here as UL (University of Lille)-FAGE (Fluorescence Assay by Gas Expansion), measured the OH reactivity in the canopy, whereas the CRM instrument, referred as LSCE-CRM, alternatively measured the OH reactivity at two heights (see Figure 1-b). Table 1 summarizes the performance of both instruments. The LP-LIF technique has a 3-fold better limit of detection than the CRM, however the CRM has a larger dynamic range since it can measure the OH

reactivity up to 300 s$^{-1}$ without sample dilution. The overall systematic uncertainty (1σ) is around 15% and 35% for the LP-LIF and the CRM, respectively. The LSCE-CRM and UL-FAGE characteristics are given in the following paragraphs.

**Table 1: Performance of the two OH reactivity instruments deployed during the LANDEX campaign.**

| Instrument | LOD*(s$^{-1}$) (3 σ) | K' max (s$^{-1}$) | Time resolution (s) | Uncertainty (1 σ) |
|---|---|---|---|---|
| LSCE-CRM | 3 | 300 | 600 | 35 % |
| UL-FAGE | 0.9 | 150** | 30-120 | 15 % |

**\* LOD: Limit of Detection; ** Without dilution**



### 2.2.1 The Comparative Reactivity Method (CRM) and instrument performance

The total OH reactivity was measured during the whole campaign, inside and above the canopy, by the LSCE-CRM instrument. This technique, first described by Sinha *et al.* (2008), is based on measuring the concentration of a reagent compound (pyrrole)

that reacts with OH under different operating conditions (i.e. steps) at the output of the sampling reactor by a PTR-MS instrument. The first step consists in introducing pyrrole with dry nitrogen and dry zero air to measure the C1 level, which corresponds to the pyrrole concentration in absence of OH. C1 accounts for potential photolysis due to photons emitted by the mercury lamp used to produce OH. During the second step, dry nitrogen and zero air are replaced by humid gases and a pyrrole concentration C2 is measured. C2 is lower than C1 because pyrrole reacts with OH. In the last step, zero air is replaced by

ambient air, which leads to a competition between the reactions of OH with pyrrole and ambient trace gases. A C3 concentration, higher than C2, is measured. The difference between C3 and C2 depends on the amount and reactivity of reactive species present in ambient air and is used to determine the total OH reactivity from eq. 2, where it is assumed that pyrrole reacts with OH following pseudo-first order reaction kinetics, i.e. [pyrrole] >> [OH]:

$$R_{OH} = \frac{(C3-C2)}{(C1-C3)} . kp . C1 \tag{Eq. 2}$$

Where *kp* is the reaction rate constant of pyrrole with OH ($1.2 \times 10^{-10}$ $cm^3$ $molecule^{-1}$ $s^{-1}$ (Atkinson, 1985)).

This technique requires multiple corrections to derive reliable measurements of total OH reactivity due to: (1) potential differences in relative humidity between C2 and C3, leading to different OH levels, (2) the spurious formation of OH in the sampling reactor when hydroperoxy radicals ($HO_2$) react with nitrogen monoxide (NO), (3) not operating the instrument under pseudo-first order conditions, and (4) dilution of ambient air inside the reactor by the addition of $N_2$ and pyrrole (Sinha *et al.*,

2008; Michoud *et al.*, 2015).

Intensive laboratory experiments as well as tests during the LANDEX field campaign were performed to characterize these corrections and assess the performances of the instrument over time. During the LANDEX field campaign, a slightly modified version of the CRM-LSCE instrument was used compared to the instrument previously deployed during the intercomparison experiment at the SAPHIR chamber (Fuchs *et al.*, 2017). Indeed, this last study showed that the OH reactivity measured by all

CRM instruments was significantly lower than the reactivity measured by the other instruments in the presence of monoterpenes and sesquiterpenes. A potential reason discussed for this discrepancy was the loss of terpenes in the inlet of the CRM instruments. The LSCE-CRM sampling system was built with ¼" OD non-heated PFA tubing and was relying on a Teflon pump to introduce the sample into the reactor. In order to measure the total OH reactivity in a monoterpene-rich environment, several technical improvements were made on the previous version of the instrument described by Zannoni *et*

*al.* (2015). First, all the PFA sampling lines were replaced by 1/8" OD sulfinert lines, continuously heated to around 50°C to prevent condensation and minimize sorption processes. Second, temperature sensors were placed at several locations inside the system to monitor potential variations; the dew point was measured in the flow out through the pump to monitor humidity fluctuations, and the pressure was also monitored to make sure that measurements were performed at atmospheric pressure.


All the flows going in and out of the reactor, the temperature at various places, the humidity and the pressure in the reactor were recorded continuously to track potential variations.

**Ambient air sampling**

Ambient air was sampled through two 1/8" OD sulfinert lines collocated on a mast close to the trailer (see Figure 1-a). The lines lengths were 8 m for the measurements performed inside the canopy and 12 m for those performed above. During sampling, the air flow was driven through one line by two pumps. The first one was a Teflon pump located upstream of the reactor and the other one was that from the Gas Calibration Unit (GCU) used to generate humid zero air from ambient

air. Together, the two pumps allowed air sampling between $1 - 1.2$ L min$^{-1}$, with the excess going to an exhaust.

**CRM-LSCE system characterization**

Several tests were performed before, during and after the campaign to assess the performance of the instrument

operated during the whole campaign. The PTR-MS was calibrated at the beginning and at the end of the field campaign showing a good stability under dry and wet conditions (slope of $15.5 \pm 0.9$ (1σ)). Regular C1 measurements were made to check the stability of the initial pyrrole concentration all along the campaign. C1 was $70.7 \pm 4.0$ (1σ) ppb.

Small differences in humidity observed between C2 and C3 were considered while processing the raw data. In order to assess this correction, experiments were performed to assess the variability of C2 on humidity by contrasting the change in C2 (ΔC2)

for various changes in the m/z 37-to-m/z 19 ratio (Δ [m/z 37-to-m/z 19 ratios]), m/z 37 and m/z 19 being representative of $H_3O^+(H_2O)$ and $H_3O^+$, respectively, and their ratio being proportional to humidity. During this campaign, three humidity tests were performed by varying the humidity in ambient air samples. These tests were in good agreement and showed a linear relationship between ΔC2 (ppb) and Δ (m/z 37-to-m/z 19 ratio) with a slope of -89.18. The correction was applied as discussed in Michoud *et al.* (2015).

An important assumption to derive $R_{OH}$ from Eq. 2 is to operate the instrument under pseudo-first-order conditions (i.e. [pyrrole]>> [OH]), which is not the case with current CRM instruments. To determine the correction factor for the deviation from pseudo-first order kinetics, injections of known concentrations of isoprene (k $_{isoprene+OH}$ = $1 \times 10^{-10}$ cm$^3$ molecule$^{-1}$ s$^{-1}$) and α-pinene (k $_{α-pinene+OH}$ = $5.33 \times 10^{-11}$ cm$^3$ molecule$^{-1}$ s$^{-1}$) (Atkinson, 1985) were performed before and after the field campaign since they represent the dominant species in this forest ecosystem. The measured OH reactivity obtained from these tests were

then compared to the expected OH reactivity, leading to a correction factor that is dependent on the pyrrole-to-OH ratio. Therefore, standard OH reactivity experiments were conducted at different pyrrole-to-OH ratios ranging from 1.7 to 4.0, which encompass the ratio observed most of the time during the campaign. These tests led to a correction factor (F) = -0.52 × (pyrrole-to-OH) +3.38.



NO mixing ratios were lower than 0.5 ppb (corresponding to the detection limit of the NOx monitor deployed during LANDEX) most of the time for the measurement time periods used in this study, and no correction was applied for the spurious formation of OH from the $HO_2+NO$ reaction. The correction (D) on the reactivity values due to the dilution was around 1.46 during the campaign. Thus, the total OH reactivity may be expressed as:

$$R_{OH\ final}=[\ \frac{(C3-C2(corrected))}{(C1-C3)}\ .\ kp\ .\ C1]\ .\ F.\ D \tag{Eq. 3}$$

Finally, overall uncertainties were estimated at 35% (1σ) for the measured OH reactivity by the CRM (Zannoni *et al.*, 2015).

### 2.2.2 UL-FAGE reactivity instrument

Total OH reactivity was measured at a different location inside canopy, from the 13[th] to the 19[th] of July, using LP-LIF instrument of the PC2A laboratory (UL-FAGE reactivity) which has already been used in several intercomparisons and field campaigns (Hansen *et al.*, 2015; Fuchs *et al.*, 2017). The reactivity instrument comprises three parts: the photolysis laser, the photolysis cell (reaction tube) and the LIF cell based on FAGE technique. The photolysis laser is used to generate OH radicals

within the photolysis cell by the photolysis of $O_3$ in the presence of water vapor. The photolysis laser is a YAG laser (Brilliant EaZy, QUANTEL) with a doubling and a quadrupling stage providing a radiation at 266 nm with a repetition rate of 1 Hz. The photolysis beam is aligned at the center of the photolysis cell and is expanded (diameter of 4 cm reaching the entrance of the cell) by two lenses (a concave one f=-25mm and a convex with f=150mm) in order to increase the photolysis volume and to limit the diffusion effect in the photolysis cell.

This photolysis cell is a stainless steel cylinder with an internal diameter of 5 cm and a length of 48 cm. It presents two openings on the opposite sides, one as an entrance for the air samples and the second connected to a pressure monitor (Keller PAA-41) to measure the pressure inside the cell. Ambient or humid clean air (which is produced by passing a fraction of dry synthetic air, purity of 99.8%, through a water bubbler, called zero air and used to determine the OH reactivity in the absence of reacting species) are injected through the first opening with a small flow of synthetic air (about 20 mL min⁻¹) passing through an ozone

generator (Scientech) to generate an ozone concentration of about 50 ppb in the total flow. The ozone concentration is chosen to produce enough OH to have a good signal/noise ratio, but kept low enough to minimize the reactions involving $O_3$.
The sampled mixture is continuously pumped into the FAGE cell (pressure=2.3 Torr) by a dry pump (Edwards, GX 600L) and the LIF signal is collected by a CPM (Perkin Elmer MP1982), an acquisition card and a LabView program. The detection of the fluorescence is synchronized with the photolysis laser pulses by delay generators. The OH reactivity time resolution was

at the minimum set to be 30 s, meaning that each OH decay was accumulated over 30 photolysis laser shots and fitted by a mono-exponential decay. The number of sets of 30 photolysis laser shots accumulated is determined according to the signal to noise ratio (S/N) obtained (typically 4). When the S/N is lower, a set of 30 OH decays is added to the previous one and so on





until reaching the criteria. As the reactivity and the humidity vary along the day, S/N varies as a function of the ambient species concentrations. In order to validate the experimental setup before the campaign, the well-known (CO + OH) reaction rate constant was measured. Different CO concentrations allowed to measure reactivities ranging from 10 to 90 $s^{-1}$ and to determine (using a linear regression: $R^2 = 0.97$) a rate constant of $k_{CO + OH} = (2.45 \pm 0.11) \times 10^{-13}$ $cm^3$ molecule$^{-1}$ $s^{-1}$, in good agreement

with the reference value of $2.31 \times 10^{-13}$ $cm^3$ molecule$^{-1}$ $s^{-1}$ (Atkinson et al., 2006) at room temperature.

**Ambient air sampling**

Ambient air was sampled in the canopy at about 5 m through a PFA line (diameter = 1/2 inches), a PFA filter being installed

at the entrance of the tube to minimize particle or dust sampling. In the photolysis cell, the gas flow was sampled at 7.5 L min$^{-1}$ and the pressure was approximately 740 Torr, i.e. lower than the atmospheric pressure due to the restriction of the flow through the Teflon sampling line. For the reactivity measurements in zero air, synthetic air from a cylinder was used and a part of the flow (2 L min$^{-1}$) passed through a bubbler filled with Milli-Q water to reach a water vapor concentration of about 3000 ppm.

**$R_{OH, zero}$ analysis**

In order to determine the OH reactivity in ambient air $R_{OH,ambient}$, it is necessary to subtract the reactivity measured using "zero air" $R_{OH,zero}$, which represents the OH losses not related to the gas phase reactions with the species of interest, present in the

ambient air, but due to wall losses, diffusion, etc, to the reactivity measured.

$$R_{OH,ambient} = R_{OH,measured} - R_{OH,zero} \qquad\qquad \text{Eq. 4}$$

Zero air tests were conducted twice a day (in the morning and at night) when the reactivity measurements took place. The average of all experiments performed with zero air leads to a mean value of $R_{OH,zero}= (4.0 \pm 0.5)$ $s^{-1}$. This value was therefore chosen as $k_{zero}$ for the whole campaign.

**2.3 Ancillary Measurements and corresponding locations**

Measurements of VOCs (Table 2) were performed at different locations (Figure 1) by a proton transfer reaction-mass spectrometer (PTR-MS) and four on-line gas chromatographic (GC) instruments. Ozone scrubbers (Copper tube impregnated with KI) and particle filters were added to the inlets of all GC sampling lines.

GC-BVOC1 is a gas chromatograph coupled to a flame ionization detector (airmoVOC C6- C12, Chromatotec), used by LSCE to monitor high-carbon VOCs (C6- C12) at 12 m height with a time resolution of 30 min. Sampling was undertaken





for 10 min. The instrument sampled ambient air with a flow rate of 60 mL min$^{-1}$. Once injected, the sample passed through a capture tube containing the adsorbent Carbotrap C, for VOCs preconcentration at room temperature; the capture tube is then heated up to 380 °C and the sample is introduced into the separating column (MXT30CE, id = 0.28 mm, length = 30 m, film thickness = 1 μm), with hydrogen as the carrier gas. During the campaign, calibrations were performed with a certified standard

containing a mixture of 16 VOCs (including 8 terpenes) at a concentration level of 2 ppb (National Physical Laboratory, Teddington, Middlesex, UK). Three calibrations were performed 3 times (at the beginning, in the middle and at the end of the campaign). As they were showing reproducible results (within 5% for all the terpenes except cineole), a mean response factor per VOC was used to calibrate the measurements. Note that limonene and cymene had close retention times which lead to overlapping peaks and for this reason, only the sum of both compounds has been reported. For further details, refer to Gros *et*

*al.* (2011). The sampling was done using a 13-m long sulfinert heated line (1/8") connected to an external pump for continuous flushing.

GC-BVOC2 is an online thermodesorber system (Markes Unity 1) coupled to a GC-FID (Agilent). It was used to monitor 20 C$_5$-C$_{15}$ BVOCs, including isoprene, α- and β-pinene, carenes, β-caryophylene at the 6 m height with a time resolution of 90 min. Ambient air was sampled at a flow rate of 20 mL min$^{-1}$ for 60 min through a sorbent trap (Carbotrap B)

held at 20°C by a Peltier cooling system. The sample was thermally desorbed at 325°C and injected into a BPX5 columns (60 m × 0.25 mm × 1 μm) using helium as carrier gas (30 min). Calibrations were performed at the beginning, in the middle and at the end of the campaign with a certified standard mixture (NPL, Teddington, Middlesex, UK, 2014) containing 33 VOCs (including 4 BVOCs: α- pinene, β-pinene, limonene and isoprene) at a concentration of 4 ppb each. The sampling was done using a 10 m long sulfinert line (1/4") heated at 55°C and connected to an external pump to adjust the sampling flow rate at 1

L min$^{-1}$.

GC-NMHC is an online GC equipped with two columns and a dual FID system (Perkin Elmer$^®$) that was described in detail elsewhere (Badol et al., 2004). It was used to monitor 65 C$_2$-C$_{14}$ non-methane hydrocarbons (NMHC), including alkanes, alkenes, alkynes and aromatics, at the 12 m height with a time resolution of 90 min. Ambient air was sampled at a flow of 15 mL min$^{-1}$ for 40 min through a Nafion membrane and through a sorbent trap (Carbotrap B and Carbosieve III) held

at -30°C by a Peltier cooling system. The trap was thermodesorbed at 300°C and the sample was introduced in the GC system. The chromatographic separation was performed using two capillary columns with a switching facility. The first column used to separate C$_6$-C$_{14}$ compounds was a CP-Sil 5 CB (50 m × 0.25 mm × 1 μm), while the second column for C$_2$-C$_5$ compounds was a plot Al$_2$O$_3$/Na$_2$SO$_4$ (50 m × 0.32 mm × 5 μm). Helium was used as carried gas. Calibrations were performed at the beginning, middle and end of the campaign with a certified standard mixture (National Physical Laboratory (NPL),

Teddington, Middlesex, UK, 2016) containing 30 VOCs at a concentration level of 4 ppb each. The sampling was done using a 13 m long sulfinert line (1/4") heated at 55°C and connected to an external pump for continuous flushing at 2 L min$^{-1}$.

GC-OVOC is an online GC-FID (Perkin Elmer$^®$) used to monitor 16 C$_3$-C$_7$ oxygenated VOCs (OVOCs), including aldehydes, ketones, alcohols, ethers, esters and six NMHCs (BVOCs and aromatics). A detailed description can be found in Roukos *et al.* (2009). The measurements were performed at the 12 m height with a time resolution of 90 min. Ambient air was





sampled at a flow rate of 15 mL min$^{-1}$ for 40 min through a water trap (cold finger, -30°C) and a quartz tube filled with Carbopack B and Carbopack X held at 12.5°C. VOCs were thermally desorbed at 280°C and injected into a CP-Lowox columns (30 m × 0.53 mm × 10 μm) using helium as carrier gas. Calibrations were performed 3 times during the campaign using a standard mixture (Apel Riemer, 2016) containing 15 compounds. This mixture was diluted with humidified zero air (RH=50%)

to reach VOC levels of 3-4 ppb. The sampling was done with the same sampling system than the GC-NMHC.

The PTR-MS (PTR-QiToFMS, IONICON Analytic GmbH) sequentially measured trace gases at 4 levels (L1=12 m, L2=10 m, L3=8 m, L4=6 m) with a cycle of 30 minutes (6 min at each level and 6 min of zero air). The drift tube was operated at a pressure of 3.8 mbar, a temperature of 70°C and a E/N ratio of 131 Td. Four identical sampling lines of 15 meters were used to sample ambient air at each height. The lines (PFA, 1/4" OD) were heated at 50°C and were constantly flushed at 10 L

min$^{-1}$ using an additional pump and rotameters. Teflon filters were used to filter particles at the entrance of the sampling lines. The PTR-MS drawn ambient air at a flow rate of 300 mL min$^{-1}$ from the different lines using Teflon solenoid valves and a 1.5-meter-long inlet (PEEK, 1/16" OD) heated at 60°C. Zero air was generated using a Gas Calibration Unit (GCU, IONICON Analytic GmbH) containing a catalytic oven and connected to L1. Ion transmissions were calibrated over the 21-147 Da mass range every 3 days using the GCU unit and a certified calibration mixture provided by IONICON (15 compounds at

approximately 1 ppm, including methanol, acetaldehyde, acetone, aromatic compounds, chlorobenzenes, etc.). Measurements of methanol, acetonitrile, acetaldehyde, acetone, isoprene, methacrolein + methylvinylketone + fragment ISOPOOH, methylethylketone, sum of monoterpenes, sum of sesquiterpenes, acetic acid, nopinone and pinonaldehyde, obtained from levels 1 and 4 corresponding to the levels where OH reactivity measurements were performed, are discussed in this article. Sesquiterpene, acetic acid, nopinone and pinonaldehyde measurements were not corrected for fragmentation in the drift tube

and we cannot rule out the detection of other isomers at these masses such as glycolaldehyde for acetic acid measurements.

Inorganic traces gases (O$_3$ and NOx) were measured by commercial analyzers deployed by IMT-Lille-Douai (L1 to L4 for O$_3$) and EPOC (L4 for NOx). The nitrate radical (NO$_3$) was measured using an IBB-CEAS instrument (Incoherent Broad Band Cavity Absorption Spectroscopy) developed by the LISA (Laboratoire Interdisciplinaire des Systèmes Atmosphériques) research group and deployed for the first time on site during the LANDEX field campaign. Meteorological

parameters such as temperature, relative humidity, global radiation, vertical turbulence, wind speed and wind direction were monitored using sensors already available at the ICOS measurement site.

**Table 2: Summary of supporting measurements performed inside and/or above the canopy**

| Instrument | Resolution time (min) | Measured species |
|---|---|---|
| GC-BVOC1 | 30 | α-pinene, β-pinene, myrcene, Δ-carene, p-cimene, limonene + cymene, cineol |
| GC-BVOC2 | 90 | α-pinene, β-pinene, myrcene, limonene, camphene, sabinene, α-phellandrene, 3-carene, p-cymene, ocimene, 1,8-cineol(=eucalyptol), α- terpinene, γ-terpinene, terpinolene |
| GC-NMHC | 30 | ethane, ethylene, propane, propene, isobutane, butane, acetylene, trans-2-butene, cis-2-butene, isopentane, pentane, 1,3-butadiene, 2-methyl-butene + 1-pentene, cyclopentene or terpene, hexene, hexane, 2,4-dimethylpentane, benzene, 3,3-dimethylpentane, 2-methylhexane, isooctane, heptane, toluene, octane, ethylbenzene, m+p-xylenes, styrene, o-xylene, nonane, 4-ethyltoluene, 2-ethyltoluene, 1,2,4-trimethylbenzene, 1,3-dichlorobenzene, undecane, isopropylbenzene, n-propylbenzene |
| GC-OVOC | 30 | furan, tert-amylmethylether, 2-butanone, ethanol, isopropanol, butanol+2-hexanone, benzaldehyde |
| PTR- MS | 6 min every 30 min at each level | methanol, acetonitrile, acetaldehyde, acetone, isoprene, methacrolein+methylvinylketone+fragment ISOPOOH, methylethylketone, sum of monoterpenes, sum of sesquiterpenes*, acetic acid*, #, nopinone*, pinonaldehyde*<br>**Fragmentation was not corrected for and reported concentrations are likely lower limits,*<br>*#potential interferences from isomeric compounds such as glycolaldehyde* |

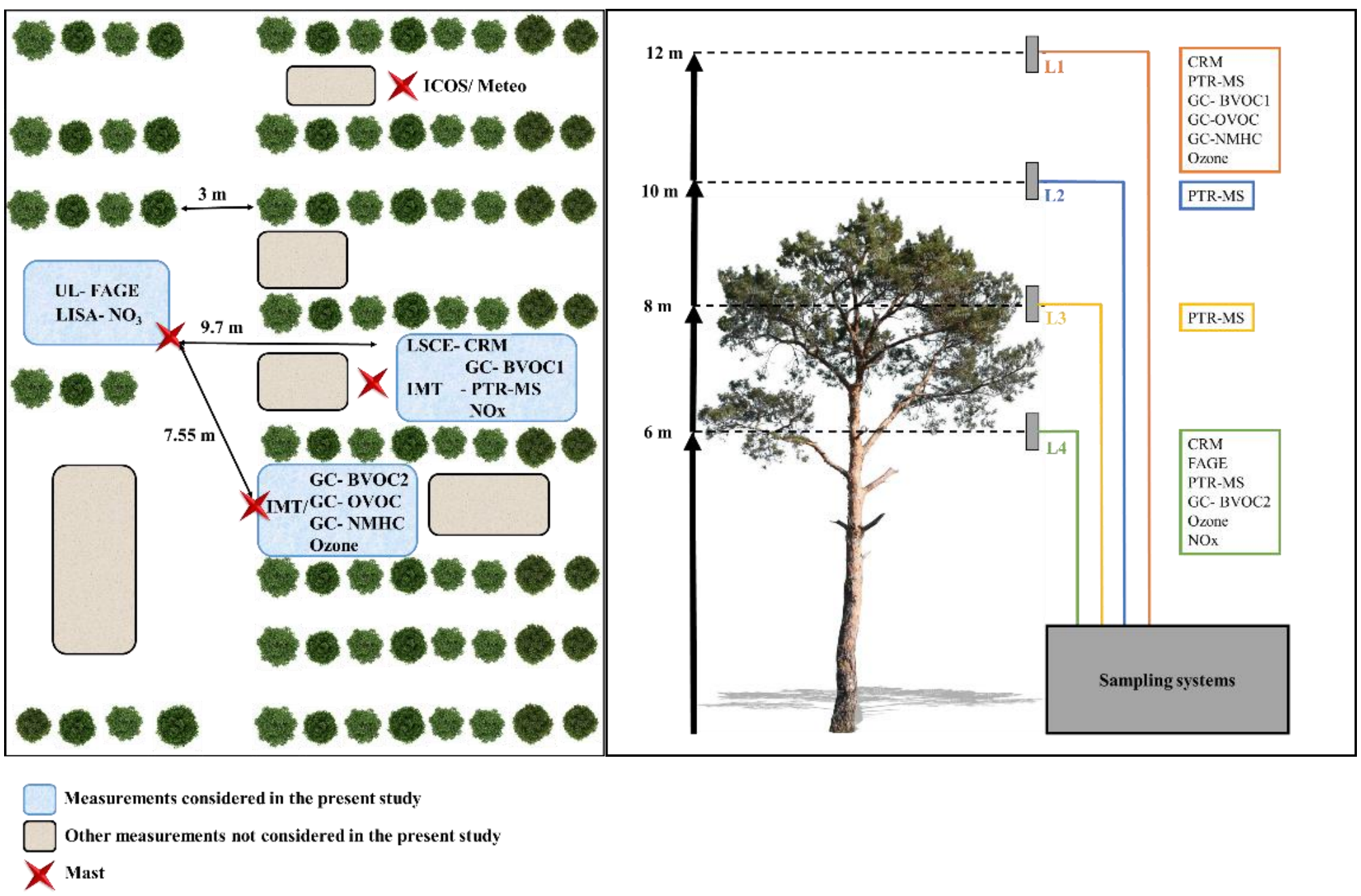

**Figure 1: Deployment of instruments at the measurement site. Left side (a) corresponds to the horizontal deployment, the right side (b) represents the different sampling levels with respect to the average trees' height.**



### 2.4 OH reactivity calculation

As different instruments were available to quantify VOCs at different locations (Figure 1 and Table 2), a selection of the data used to calculate the OH reactivity (Eq. 1) was made, based on data availability for the different instruments (Table S1).

Since measurements from the PTR-MS instrument covers the whole campaign and were performed at the same heights than OH reactivity measurements, these measurements, including methanol, acetonitrile, acetaldehyde, acetone, isoprene, methacrolein+methylvinylketone+fragment ISOPOOH, methylethylketone and the sum of monoterpenes, were selected to calculate the OH reactivity and to evaluate the potential missing OH reactivity at both levels. However, using only this set of data presents some limitations:

1)    The PTR-MS only measures the sum of monoterpenes (m/z 137+ m/z 81), while the detected monoterpenes are speciated by the GCs.

2)    It was observed that isoprene measurements at m/z 69 were disturbed by the fragmentation of some terpenic species (Tani, 2013; Kari et al., 2018), which led to a significant impact on the night-time measurements when isoprene was low.

3)    Some NMHCs and OVOCs measured by GC at the 12 m height were not measured by the PTR-MS. This requires to
assess the contribution of these additional species to the total OH reactivity for both heights.

To overcome these limitations, several tests were made to evaluate the reliability of the PTR-MS data to calculate the OH reactivity.

1)    In order to use the sum of monoterpenes measured by the PTR-MS to calculate the total OH reactivity, it was necessary to determine a weighted rate constant for the reaction of monoterpenes with OH. After checking the consistency between the two GCs (BVOC1 and BVOC2, see supplementary material S2) and comparing the sum of monoterpenes measured by each GC to the PTR-MS measurements (simultaneous measurements at the same height - Fig. S2b and c), the weighted rate constant was calculated as the sum of the rate constants of each OH + monoterpene reaction multiplied by the average
contribution of each specific monoterpene to the sum. The contribution of each monoterpene was calculated by dividing the concentration of the 8 speciated monoterpenes that were measured by both GCs (α-pinene, β-pinene, myrcene, Δ-carene, p-cimene, limonene + cymene, cineol), by their total concentration (Fig. S3a). The weighted rate constant is defined as:

$$k_{OH,weighted} = \sum_i k_{OH+ X_i} \, F_i$$

**Eq. 5**

Where $F_i$ represents the contribution of each individual species to the total concentration of monoterpenes, and $k_{OH+ X_i}$ the
corresponding rate constant with OH. The reaction rate constant of the different trace species quantified in the field were taken



from the literature (Atkinson et al., 2006). The OH reactivity of monoterpenes measured by PTR-FMS was calculated according to the following equation:

$$R_{OH-monoterpenes} = k_{OH,weighted} \times [MT]$$   **Eq. 6**

where [MT] represents the sum of monoterpenes measured by PTR-MS.

The calculated OH reactivity inside and above the canopy (Fig. S3b and e) from (i) the use of the weighted OH reaction rate constant and the total concentration of monoterpenes measured by GC and (ii) the use of individual species and their associated rate constants are in relatively good agreement as shown by the scatter plots. A slope of 0.95, $R^2$=0.99 has been obtained using the monoterpenes measured with the GC-BVOC1 at 12 m (Fig. S3c); a slope of 0.94, $R^2$=1.0 using the same 8 compounds commonly monitored with GC-BVOC1 but measured at 6 m with GC-BVOC2 (Fig. S3f)). When replacing the total

concentration of monoterpenes measured by GCs by the PTR-MS measurements, slopes of 1.22 and 1.19 were obtained at 12 and 6 m heights, respectively (Fig. S3d and S3g). This increase in the slope values is likely due to an underestimation of the total monoterpene concentration by the GC instruments since these instruments only measured the most abundant monoterpenes present at the site. We cannot rule out a small overestimation of monoterpenes by the PTR-MS since fragments from other species such as sesquiterpenes could be detected at the monoterpene m/z. However, this interference should be

negligible due to the low concentration of ambient sesquiterpenes. These results are in agreement with the scatterplots comparing the sum of monoterpenes measured by GC and by PTR-MS (slopes of 1.29 and 1.10 at the 12 and 6 m heights, respectively, see fig. S2b and S2c). Thus, the PTR-MS data was used to calculate the OH reactivity from monoterpenes for both heights, with a weighted reaction rate constant of $76 \times 10^{-12}$ cm$^3$ molecule $^{-1}$ s$^{-1}$ at the 12 m height and $77.9 \times 10^{-12}$ cm$^3$ molecule $^{-1}$ s$^{-1}$ at the 6 m height.

2)   As mentioned above, some monoterpenes have been observed to fragment at m/z 69.0704, which would result in an interference for isoprene measurements. In order to use the PTR-MS data for this species (only instrument measuring isoprene at 12 m), the contribution of monoterpenes to m/z 69 has been estimated by comparing the GC-BVOC2 and PTR-MS measurements of isoprene performed at 6 m. This comparison showed that approximately 4% of the monoterpene concentration measured by PTR-MS had to be subtracted to that measured at m/z 69.0704 to get a good agreement between

the PTR-MS and GC-BVOC2 measurements of isoprene as shown in Fig. S4a.

3)   A large range of NMHCs and OVOCs were measured at the 12 m height only by GC-NMHC and GC-OVOC (Table 2). Butanol (from SMPS exhausts) was also checked and found to be negligible at 12 m and highly and rapidly variable at 6 m (short peaks). Thus, it was chosen not to take these species into account in the OH reactivity calculations. However, sensitivity tests were performed, in order to compute the relative contribution of butanol, OVOCs and NMHCs to OH

reactivity (See section 3.5 and Fig. S5 and S6). Regarding methane and carbon monoxide, an estimation was made seen





their relatively low k reaction rate coefficient with OH, taking mean concentration values of 2000 ppb and 150 ppb, respectively.

The above limitations are summarized in Table S7 (supplementary material). Data used to calculate the OH reactivity has been resampled to 1 min, based on a linear interpolation (see Table 2 for the respective time resolution of the different
5 instruments). This time base was chosen to be comparable to the time resolution of the UL-FAGE reactivity instrument, in order to keep the dynamics in OH reactivity variability.

## 3. Results

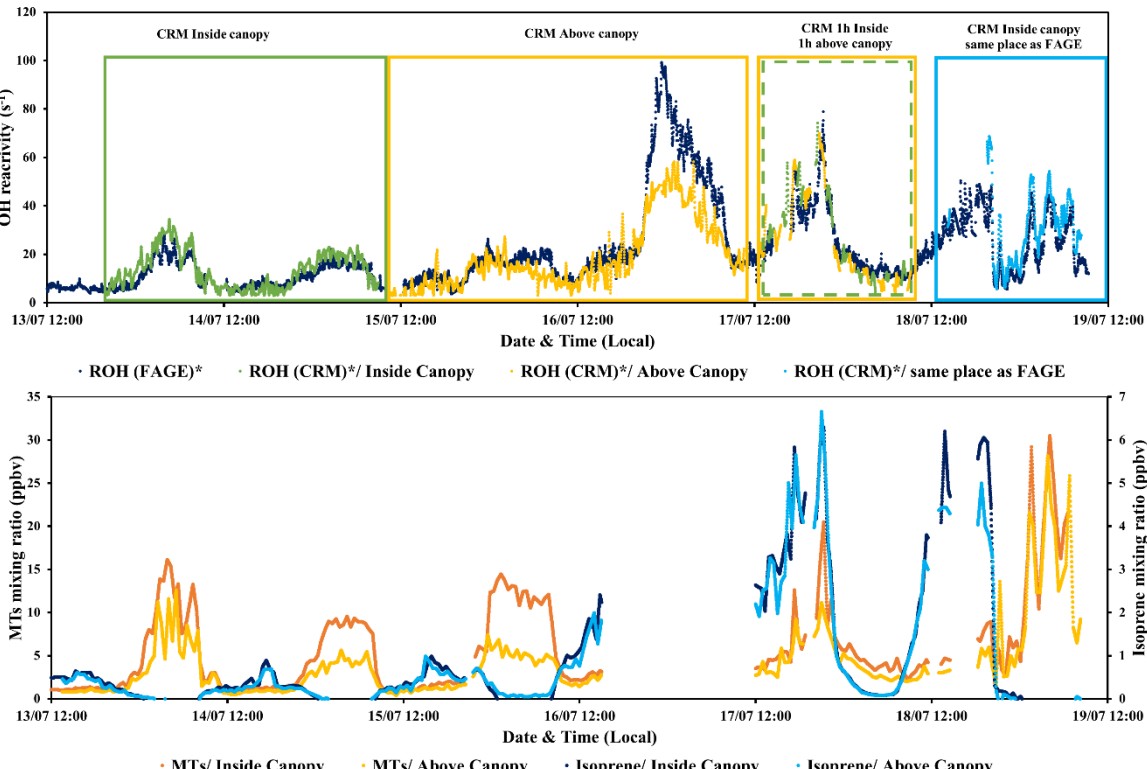

**Figure 2: a) Time series of total OH reactivity measured by UL-FAGE and LSCE-CRM instruments from the 13th to 19th of July 2017 (upper graph). Dark blue symbols represent the measured reactivity by UL-FAGE, green, yellow and blue symbols represent the measured reactivity by LSCE-CRM inside canopy, above canopy and inside canopy at the same location as the UL-FAGE instrument, respectively. The lower graph (b) shows the measured monoterpenes (MTs) and isoprene in the field for the same period. Dark blue and light blue dots correspond to isoprene concentrations at 6 and 12 m height, respectively. Orange and yellow dots represent monoterpenes concentrations at 6 and 12 m height, respectively.**



Measurements performed by both instruments at the same location were first compared to evaluate the agreement between the two techniques. The horizontal variability of total OH reactivity (same height) is also discussed. A second part of the result section is dedicated to a description of the total OH reactivity variability on the vertical scale with some

meteorological parameters. A comparison between measured and calculated OH reactivity for both the 6 and 12 m heights as well as a description of the BVOC contributions to the measured OH reactivity are then presented. Finally, we discuss the missing OH reactivity observed during this campaign and its possible origin.

### 3.1.1 Inter-comparison of LSCE-CRM and UL-FAGE OH reactivity measurements at the same location

The direct comparison between LSCE-CRM and UL-FAGE reactivity instruments was done during the last two days of the campaign. The sampling line of LSCE-CRM was moved to be collocated to the sampling line of UL-FAGE. Both instruments were measuring at the same location inside the canopy level, above the UL container at 5 m height. In this way the comparison between both instruments was made possible while minimizing the variabilities which could be related to the heterogeneity in ambient air. During this period, similar values were measured by both instruments, as shown in figure 2a

(blue frame), with total OH reactivity ranging between 5 and 69 s$^{-1}$. The lowest values were observed during day-time.

When the OH reactivity measured by LSCE-CRM is plotted as function of the OH reactivity measured by UL- FAGE (Fig. 3), the linear fit exhibits a slope of 1.17± 0.02, an intercept of 4.2 ± 0.4 s$^{-1}$ and a R$^2$ of 0.87. This high intercept is statistically significant and can be due to an overestimation of the UL-FAGE reactivity instrument zero, related to the quality of zero air used for zeroing the instrument. Indeed, previous comparisons have shown that using zero air of better quality

(99.999%) may result in a zero of about 2 s$^{-1}$ lower (Hansen et al., 2015). An intercomparison of OH reactivity instruments made in the SAPHIR chamber (Fuchs *et al.*, 2017) has also shown a positive bias of 1 s$^{-1}$ for the UL-FAGE instrument when high grade zero air was flushed in the chamber. A maximum overestimation of the UL-FAGE zero by 3s$^{-1}$ is possible for this study. However, the observed intercept could also be due to an offset in the LSCE-CRM measurements, that could be related to a possible desorption of "sticky" compounds from the Teflon pump. These results indicate that both instruments respond

similarly (within 20%) to changes in OH reactivity and the offset of 4.2 s$^{-1}$ has to be taken into account when OH reactivity measurements from LSCE-CRM and UL-FAGE are further compared for different locations and heights. It is worth noting that the higher points of OH reactivity observed in figure 3 correspond to the period from 19h30 to 20h (local time) of the 18$^{th}$, July when the ambient relative humidity increased quickly by 20% which was not seen on previous days and may have interfered with LSCE-CRM OH reactivity measurements.





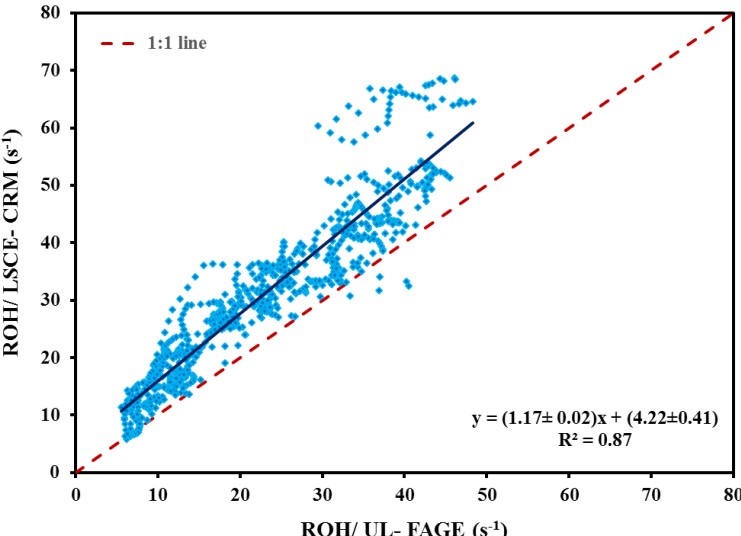

**Figure 3: Measured reactivity by LSCE- CRM instrument as function of the measured reactivity by UL-FAGE when both instruments were measuring at the same location within the canopy (data resampled with a time resolution of 1 min).**

### 3.1.2 LSCE-CRM and UL-FAGE OH reactivity measurements at two different locations inside the canopy

From the 13[th] to 15[th] midday of July (1[st] period) and from the 17[th] midday to 18[th] midday (2[nd] period), the two instruments were sampling from different locations within the canopy (with sequential within/above canopy measurements for CRM during the second period). The horizontal distance between the two inlets was around 10 m as shown in figure 1. Similar trends in OH reactivity are seen between the two datasets, even if the first period was associated with a clear vertical stratification (green frame), leading to higher concentrations of monoterpenes within the canopy, whereas the second period was characterized by a higher vertical mixing (green/ yellow frame), leading to similar concentrations of monoterpenes at the two heights. These observations are linked to the vertical turbulence which influence BVOC levels inside and above the canopy, resulting in a more or less important vertical stratification, as discussed in section 3.2.

The linear regression of LSCE- CRM data plotted against UL-FAGE data (not shown) indicates a good agreement with a slope of $1.22 \pm 0.01$, an intercept of $-0.69 \pm 0.17$ and a correlation coefficient of 0.85 (1[st] and 2[nd] period). This change in the intercept between this analysis and the analysis done when both instruments were measuring at the same location, is likely due to air mass inhomogeneities. From these observations, we can conclude that reactivity measurements performed at different horizontal locations are consistent and that inhomogeneities in ambient air can lead to differences on the order of several $s^{-1}$.



## 3.2 Measured OH reactivity and meteorological parameters

Figures 4a and 4b show the variability of total OH reactivity measured inside and above the canopy by LSCE-CRM and UL-FAGE, together with global radiation, temperature and friction velocity. Considering the whole campaign, the measured OH reactivity at both heights shows a diurnal trend ranging between LOD (3 $s^{-1}$) and 99.4 $s^{-1}$ inside canopy and

between LOD and 70.2 $s^{-1}$ above canopy, with maximum values of OH reactivity mostly recorded during nights. These OH reactivity levels are larger than other measurements performed in forested environments (Lou et al., 2010), with maximum values of approximately 80 $s^{-1}$ reported for the tropical forest (Edwards *et al.*, 2013).

The predominant meteorological parameter that had a role on OH reactivity levels was the friction velocity. It traduces the vertical turbulence intensity that was high during the day (mean day-time u* ≥ 0.4 m/s) and lower during most nights (mean

night-time u* ≤ 0.2 m/s). Based on this parameter, night-time OH reactivity (between 21:00 and 06:00 local time of the next day) was separated in 3 classes:

- Class S: Stable atmospheric conditions (mean u* ≤ 0.2 m/s)
- Class U: Unstable atmospheric conditions (mean u* ≥ 0.4 m/s)
- Class SU: Stable and unstable conditions during the same night.

The lower vertical turbulence intensity, observed for "S" nights as well as for some hours of "SU" nights, led to a lower boundary layer (Saraiva and Krusche, 2013) and a significant nocturnal stratification within the canopy, with higher concentrations of primary compounds within the canopy (Fig. 4c). These stable atmospheric conditions, together with no photochemical oxidation of BVOCs, resulted in higher total OH reactivity during these nights due to higher BVOCs concentration even though their emissions are lower compared to day-time (Simon et al., 1994).

Another important parameter to consider is ambient temperature, which enhances BVOC emissions during the day when stomata are open, and which also plays a role for night-time emissions due to permeation, even though stomata are closed in the dark (Simon et al., 1994). Considering temperature, 2 sub-classes can be added to night-time OH reactivity classification: the sub-class "Wn" corresponding to warm nights (nights with mean T ≥ 18.9°C which is the mean night-time temperature over the whole campaign) and the sub-class "Cn" that includes cooler nights (nights with mean T < 18.9°C). Thus, comparing

"S/Wn" nights and "S/Cn" nights, it can be seen that, for similar turbulent conditions, the magnitude of the measured OH reactivity was temperature dependent. Indeed, higher OH reactivity values were linked to higher ambient temperatures: nights of the 4th-5th, 6th-7th and 16th-17th of July (S/Wn) were characterized by an average temperature of 21°C compared to 16.6°C for the nights with lower OH reactivity (S/Cn).







**Figure 4: Variability of measured OH reactivity by LSCE-CRM and UL- FAGE, inside and above the canopy with a) global radiation (black), b) temperature (red), friction velocity (green) and with c) monoterpenes and isoprene concentrations. Yellow stripes indicate stable night-time atmospheric conditions (S nights with mean u* ≤ 0.2 m/s) and blue stripes indicate unstable night-time conditions (U nights with mean u* ≥ 0.4 m/s). Class SU includes nights with stable and unstable atmospheric conditions (blue + yellow stripes). Wn and Wd stand for warm nights and warm days respectively. Cn and Cd stand for cooler nights and cooler days respectively. Red dashes and black dashes indicate the temperature thresholds to distinguish warm and cool days and nights, respectively. Green dashes indicate the friction velocity threshold to distinguish stable and unstable nights.**



Regarding the period when measurements were done simultaneously at both heights (15th to 18th of July, LSCE-CRM above canopy and UL-FAGE within canopy), we can analyze the effect of turbulence on the above-within canopy differences keeping in mind a potential instrumental offset of a few s$^{-1}$ between the two methods (section 3.1). For the night of the 16th-17th of July (S/Wn) when the vertical turbulence was relatively low, total OH reactivity measured above the canopy (LSCE-CRM) was lower than the one measured inside the canopy by a mean factor of 1.6 (UL-FAGE reactivity) despite similar general trends. For the night of the 17th-18th of July (SU/Wn), stable atmospheric conditions started to settle at the beginning of the night (20h30 local time) inducing a similar stratification to that observed on the previous nights. However, this situation did not last the whole night since these stable conditions were disturbed by higher turbulences around 21h00. This led to a decrease in OH reactivity values going to similar levels inside and above the canopy (Table 3). A similar event occurred during the night of the 18th-19th of July, where three OH reactivity peaks showed up, not correlated neither with variation of turbulence intensity nor with temperature changes. However, it is worth noting that this night was characterized by an intense wind, rain and thunders which could have led to the observed bursts of BVOCs (Nakashima et al., 2013), leading to distinct peaks of BVOCs and total OH reactivity and thus relatively high total OH reactivity compared to other nights from the same class.

Total OH reactivity also increased during the day, although on a lower extent than during night-time, and reached a day-time maximum of up to 74.2 s$^{-1}$ inside the canopy and 69.9 s$^{-1}$ above the canopy, following the same trends than temperature and solar radiation. Temperature appeared to be an important driving factor of total OH reactivity during day-time hours, therefore, day-time OH reactivity was divided into 2 classes: Class "Wd" with warm conditions (mean daytime T ≥ 24°C) and class "Cd" with cooler temperatures (mean day-time T < 24°C) indicated on figure 4. The solar irradiation also played a role on day-time OH reactivity since it is responsible of initiating the emission of some compounds like isoprene, that is light and temperature dependent. Thus with the first rays of sunlight, the emission and the concentration of isoprene increased leading to an increase in total OH reactivity.

Examining BVOC profiles (Fig. 4-c), we can see how the variability of primary BVOC concentrations can explain the day/night variability of total OH reactivity. Indeed, monoterpenes, which are the main emitted compounds in this ecosystem, were influenced by vertical turbulence and night-time temperature, exhibiting a diurnal profile with maxima during stable nights and minima during day-time. Under stable atmospheric conditions (class S), monoterpenes concentration started to increase at the beginning of the night (between 20h and 21h local time) corresponding to the time of the day when the turbulence intensity started to drop and the nocturnal boundary layer started to build up. Maximum mixing ratios were reached in the middle of the night, corresponding to a lower dilution in the atmosphere and a lower oxidation rate (low OH concentrations, nitrate radical mixing ratios lower than the LOD (3ppt/min) most of the time, and BVOC's chemistry with ozone generally slower than during daytime (Ciccioli et al., 2002)). Finally, the monoterpenes concentration dropped as soon as the first sunlight radiations broke the stable nocturnal boundary layer inducing lower levels of OH reactivity. Under these conditions, the concentration of monoterpenes inside the canopy was higher than above the canopy, showing a clear stratification, consistent with differences seen on total OH reactivity at the different heights. On the contrary, during turbulent




night hours (Class U and SU), the concentration of monoterpenes was lower inside the canopy and similar to that observed above, leading to lower and closer night-time OH reactivity at both measurements heights.

At the end, even though BVOC emissions are more intense during the day (Simon et al., 1994), the higher turbulence observed compared to night-time led to a faster mixing within the canopy and thus similar levels of isoprene and monoterpenes inside and above the canopy. These day-time levels were lower than those observed at night for monoterpenes and higher for isoprene, the latter being light and temperature dependent.

To conclude, these observations show that on one hand, lower turbulence inducing stable atmospheric conditions during the night explains the observed stratification in terms of monoterpenes levels and thus in terms of OH reactivity levels within the canopy, when on the other hand, higher turbulence during day-time leads to higher mixing within the canopy and a vertical homogeneity, with similar BVOCs concentrations and OH reactivity levels at both heights.

**Table 3: Classification of night-time OH reactivity measured by LSCE-CRM based on the mean vertical turbulence intensity (u\* m/s) and the night-time mean temperature (°C). \* refers to measurements with UL-FAGE, inside the canopy.**

| Class | Nights | Mean u* (m/s) | Mean T (°C) | Sub-class | Total OH reactivity (s⁻¹) | |
|---|---|---|---|---|---|---|
| | | | | | Inside Canopy | Above Canopy |
| S | 14th-15th, july | 0.1 | 15.2 | Cn | 17.1 - 13.7* | --- |
| | 3rd- 4th, July | 0.1 | 15.6 | | 25.7 | --- |
| | 15th-16th, July | 0.1 | 16.6 | | 17.5* | 15.6 |
| | 11th- 12th, July | 0.1 | 17 | | --- | 15.8 |
| | 10th- 11th, July | 0.1 | 17.3 | | --- | 13.0 |
| | 13th-14th, July | 0.1 | 18 | | 18.7 - 13.4* | --- |
| | 6th- 7th, july | 0.1 | 20.1 | Wn | 37.5 | --- |
| | 16th- 17th, july | 0.1 | 21.6 | | 67.2* | 43.2 |
| | 4th- 5th, july | 0.2 | 21.7 | | 45.6 | --- |
| U | 8th- 9th, july | 0.5 | 19.5 | Cn | 7.3 | --- |
| SU | 5th- 6th, July | 0.3 | 18.9 | Wn | 12.6 | --- |
| | 18th- 19th, july | 0.3 | 20.1 | | 28.8 - 21.5* | --- |
| | 7th- 8th, july | 0.4 | 21.2 | | 18.7 | --- |
| | 17th- 18th, july | 0.3 | 23.2 | | 20.4 - 19.6* | 20.5 |

**3.3 Measured and calculated $R_{OH}$ within and above the canopy**

Figure 5 shows that there is a good co-variation of the measured total OH reactivity by the CRM instrument with the values calculated from the PTR-MS data. However, a certain fraction of the measured total OH reactivity remains unexplained by the considered compounds (Table 2). Diurnal variations of OH reactivity were observed within the canopy, during the major part of the campaign, with maximum values recorded during most nights and averages of 19.1 ± 12.7 s⁻¹ and 19.3 ± 16.3 s⁻¹ measured by the LSCE-CRM and UL-FAGE instruments, respectively. This diurnal cycle was also observed above canopy





where the average total OH reactivity was $16.5 \pm 12.3$ s$^{-1}$, which is higher than observations made in other temperate coniferous forests (Ramasamy et al., 2016) where the reported OH reactivity ranges from 4-13 s$^{-1}$ (campaign average).

During the first part of the campaign (3$^{rd}$ – 10$^{th}$ of July), when the LSCE-CRM was measuring alone inside the canopy, total OH reactivity varied between LOD (3 s$^{-1}$ at 3σ) and 76.9 s$^{-1}$, while the calculated reactivity ranged between 1.8 and 60 s$^{-1}$. During the second period (13$^{th}$ – 15$^{th}$ and 17$^{th}$ – 18$^{th}$ of July), similar maxima were recorded by the LSCE- CRM (74.2 s$^{-1}$) and the UL- FAGE instruments (78.9 s$^{-1}$), when both were measuring at two different locations within the canopy. Regarding the calculated OH reactivity, it varied between 2.6 and 59.3 s$^{-1}$. During this same period, the FAGE instrument measured alone within the canopy from the 15$^{th}$ to the 17$^{th}$ of July and recorded total OH reactivity values ranging between 3.6 and 99.2 s$^{-1}$, however the PTR-MS data were not taken into account for the period going from the 16$^{th}$ 15:00 to the 17$^{th}$ 12:00 due to an electrical failure. Finally, during the last two days (18$^{th}$- 19$^{th}$ of July), total OH reactivity showed a particular behavior as mentioned in section 3.2. It started to increase in the afternoon, reached a maximum at the beginning of the night that was suddenly broken by turbulences and showed three peaks during the night corresponding to more stable conditions observed for both the measured and calculated reactivity.

Regarding above canopy measurements, the measured OH reactivity varied between LOD and 35.7 s$^{-1}$ between the 10$^{th}$ and the 12$^{th}$ of July, whereas the calculated reactivity varied between 1.2 and 14.5 s$^{-1}$. A similar trend was observed for the second period of measurements performed above the canopy (15$^{th}$ - 18$^{th}$ of July) during which higher OH reactivity was recorded with a maximum of 69.9 s$^{-1}$, which is 1.7 times higher than the calculated OH reactivity (40.8 s$^{-1}$).

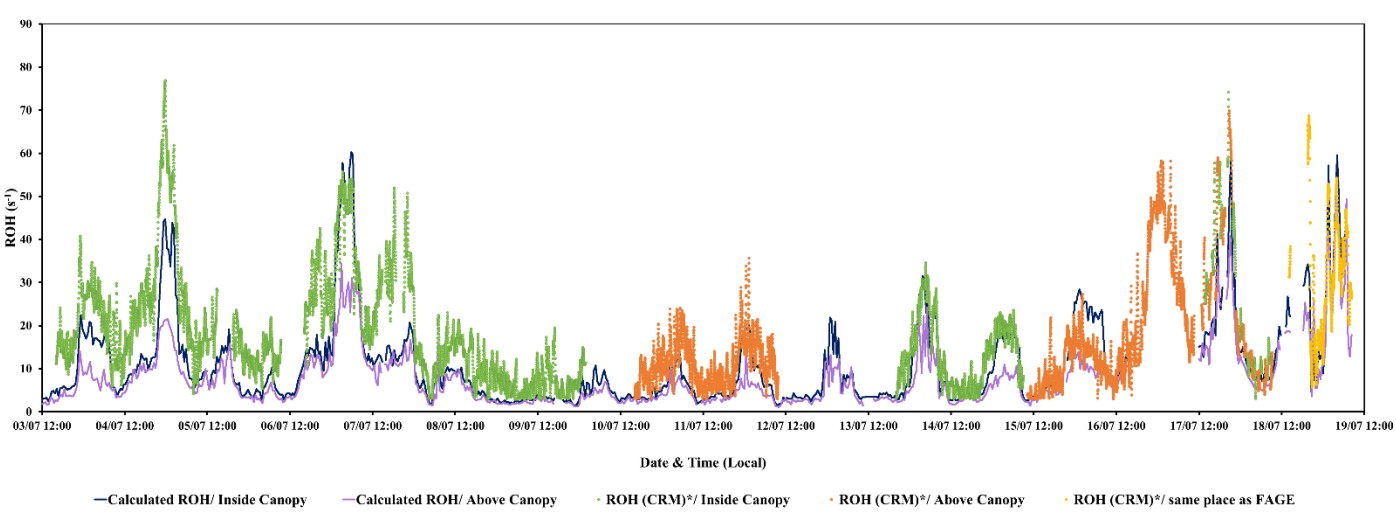

**Figure 5: Variability of measured ROH (LSCE-CRM) and calculated ROH (PTR-MS) at 6 and 12 m height.**





### 3.4. Contribution of VOCs (PTR-MS) to calculated OH reactivity within and above the canopy

Figure 6 shows the breakdown of trace gases to the calculated OH reactivity during day-time and night-time at the two heights, taking into account the whole measurement period (campaign average). We note that primary BVOCs (monoterpenes, isoprene) are by far the main contributors to the calculated OH reactivity, representing 93- 97 % of the calculated OH reactivity on average.

Monoterpenes exhibited the most prominent contribution to the calculated OH reactivity. These species had a similar contribution within and above the canopy, but significant differences between day-time (63-65%) and night-time (91-88%). Next to monoterpenes, isoprene had a maximum contribution during day-time and represented on average 28-30% of the calculated OH reactivity, followed by acetaldehyde (3%) and MACR + MVK (3%) at both measurements heights. However, during night-time, isoprene accounted for only 5-7% of the OH reactivity measured within and above the canopy, acetaldehyde contributing for approximately 2% and MACR + MVK less than 1.5%.

Thus, we can conclude that no substantial difference in the atmospheric chemical composition existed between the two sampling heights, even when we only consider stable nights (monoterpenes relative contribution is around 92% inside and above the canopy).

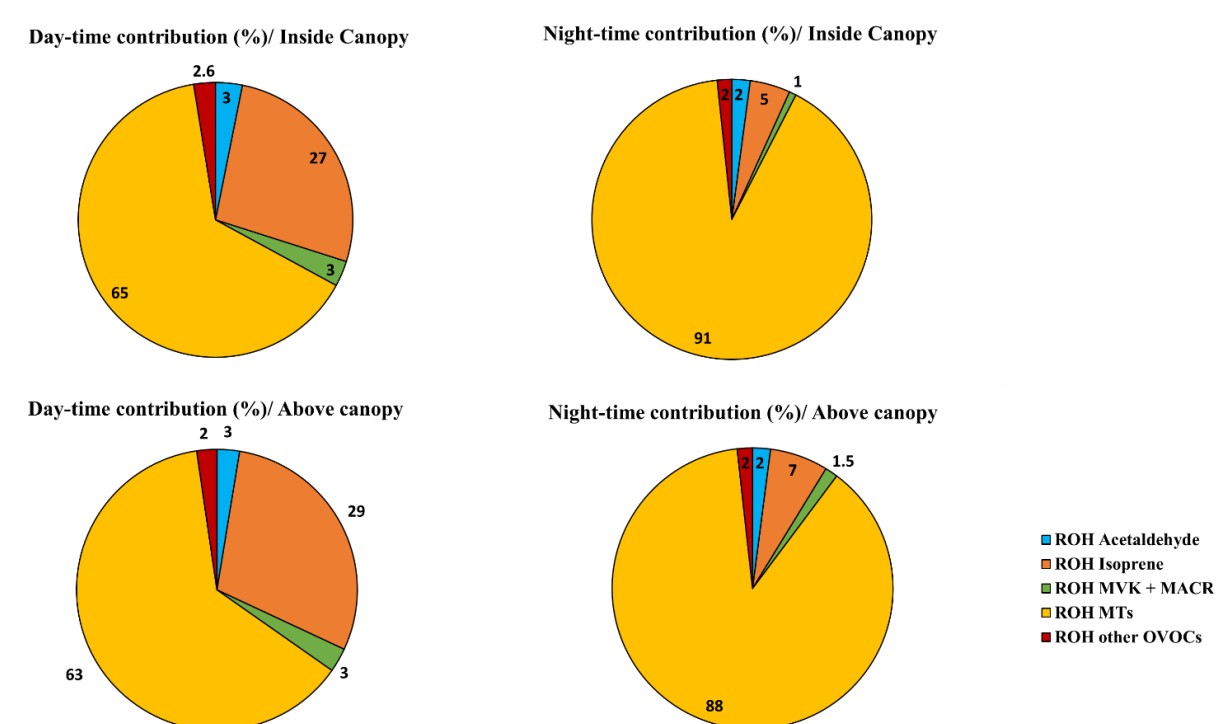

**Figure 6: The components of calculated OH reactivity within and above the canopy during day-time and night-time. Total OH reactivity was 18.3 s⁻¹ and 21.9 s⁻¹ (mean values) at 6m height and 12.2 s⁻¹ and 21.6 s⁻¹ at 12 m height during the day and the night, respectively. The compounds measured by the PTR-MS (table 2) were used to calculate their relative contributions.**





### 3.5 Description and investigation of potential missing OH reactivity during the LANDEX campaign

The missing OH reactivity was calculated as a difference between the total OH reactivity measured by LSCE-CRM, since it was operated over the whole campaign and at both heights, and the OH reactivity calculated from PTR-MS data. It is worth noting that a scatter plot of the LSCE-CRM and UL-FAGE data led to a slope of 1.2 and an intercept of 4.2 s⁻¹ (section

3.1.1), leading to higher values for the CRM instrument. The intercept is mainly attributed to a zeroing issue on UL-FAGE but we cannot rule out a bias on the CRM measurements. Considering OH reactivity values measured by the CRM instrument may therefore maximize the missing OH reactivity if the discrepancy observed between the two instruments is due to a bias in the CRM data. In the following, the analysis on the missing OH reactivity was performed when it was higher than both the LOD of 3 s⁻¹ (3σ) and 35% of the measured OH reactivity (uncertainty on the CRM measurements).

Figure 7 shows a) the variability of the missing OH reactivity within and above the canopy, together with ambient temperature, b) friction velocity (red), and ozone mixing ratios within (yellow) and above (blue) the canopy. The ozone mixing ratios variability is analyzed as ozone chemistry can dominate night-time chemistry regarding the dominating BVOCs on this site (a-pinene, b-pinene)(Ciccioli et al., 2002; Kammer et al., 2018). It showed a diurnal cycle with maximum values during the day that were similar within and above the canopy due to the high vertical turbulence, and lower levels during stable nights

(class S) with generally slightly higher levels above the canopy. On an average campaign, the relative missing OH reactivity was 37.4 % and 47.8% inside and above the canopy, respectively, which is lower than observations performed in the tropics and subtropics (50-70 %) and higher than values reported in other temperate forests (30 %) (Nakashima et al., 2013). However, as shown in figure 7, the missing OH reactivity showed high day/night and inside/above canopy variabilities, which is discussed in details below.

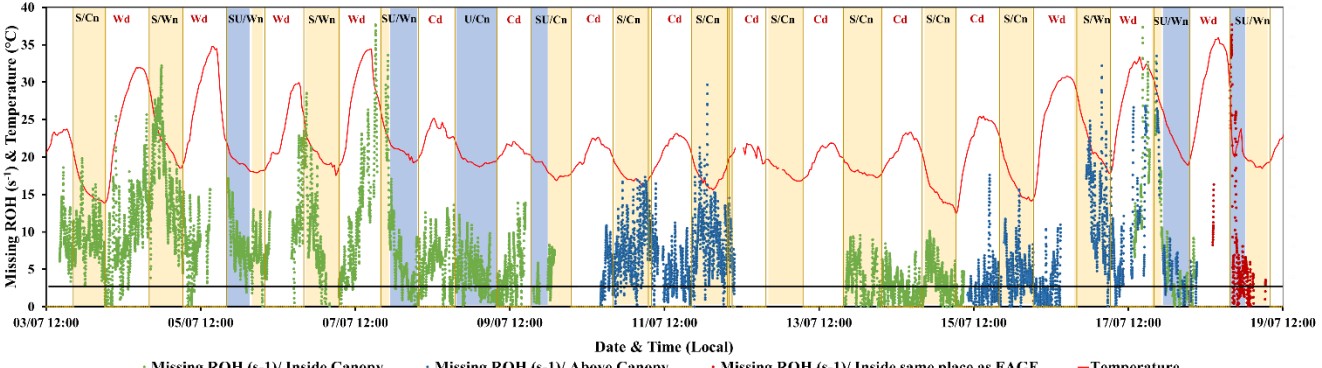









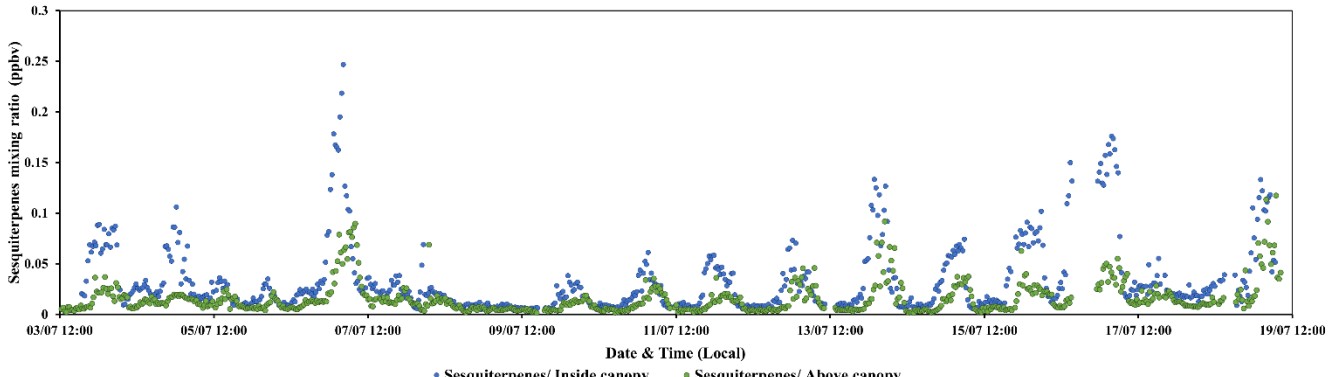

**Figure 7: Missing OH reactivity inside and above the canopy together with a) temperature, b) friction velocity (red), ozone mixing ratios inside (yellow) and above (blue) the canopy, c) relative humidity (clear blue), MACR+ MVK (dark blue) and acetic acid (green) inside the canopy, d) Nopinone (yellow) and pinonaldehyde (purple) inside the canopy and e) sesquiterpenes inside (blue) and above (green) the canopy.**

**Missing OH reactivity inside the canopy:** At 6 m height, a fraction of 41.3±15.5 % (6.9± 4.6 s$^{-1}$) of the measured OH reactivity remained unexplained during the day (campaign average) when comparing it with the calculated OH reactivity.

Analyzing the behavior of the missing reactivity during day-time, Figure 8 shows that it increases exponentially with temperature. Indeed, the average missing OH reactivity was around 9.5 s$^{-1}$ for "Wd" days, which is 2.2 times higher than for "Cd" days. As reported in Di Carlo et al. (2004), the missing OH reactivity was fitted with an equation usually used to describe temperature-dependent emissions of monoterpenes (Guenther et al., 1993): $E(T) = E(293)exp(\beta(T-293))$, where E(T) represents the emission rate at a given temperature T. In this equation, E(T) was substituted to MROH(T) and E(293) by

MROH(293) with MROH representing the missing OH reactivity (Hansen et al., 2014). The value of β determined from the fit of the data for the 6 m height (day-time), is about 0.09, consistent with the value obtained during the PROPHET 2000 (Di Carlo et al., 2004) and in the range of typical β values (0.057 to 0.144 K$^{-1}$), showing that the temperature dependence of the missing OH reactivity was close to that observed for terpene emissions. This observation was a clear evidence that the missing OH reactivity came from unmeasured terpene-like, temperature dependent BVOCs, which can also be the case in our forest

ecosystem. Nevertheless, we cannot exclude the possibility of light and temperature dependent emissions. Indeed, Kaiser *et al.* (2016) also investigated the temperature dependency of day-time missing OH reactivity in an isoprene-dominated forest, reporting that part of the missing emissions could be characterized by a light and temperature dependence, knowing that temperature increases with increasing solar radiation.



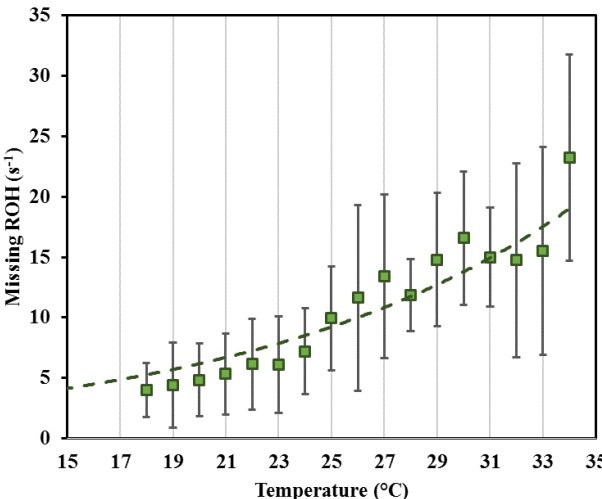

**Figure 8: Day-time missing OH reactivity binned by ambient temperature for the 6m height for temperatures ranging from 15-35 °C. Error bars represent the standard deviation on average missing OH reactivity calculated for each temperature bin.**

Regarding night-time hours, the highest mean missing OH reactivity was seen for the S/Wn of 4[th]-5[th] of July (17.5 s[-1]). This observation shows that lower vertical turbulence and higher temperatures could be favorable conditions for night-time unmeasured emissions and/ or secondary products formation. However, despite that the night of 6[th]- 7[th] of July is a S/Wn night too, the missing OH reactivity was lower than the mean missing OH reactivity observed on the 4[th]- 5[th] of July night, indicating that other processes influence the missing OH reactivity during the night.

Considering other measurements performed on the site, not included in the calculation, such as NO, $NO_2$ and ozone measurements within the canopy (6 m), and assuming constant concentration of CO and methane, it showed that their contribution can reach on average 1.5 s[-1] (maximum around 3.1 s[-1]) at this level, explaining between 4 and 20% of the observed missing OH reactivity. However, the missing OH reactivity was still significant for "Wd" days (35% of the measured OH reactivity), "S/Wn" nights (34% for the 4[th]-5[th], July night) and "SU/Wn" nights (39%).

**Missing OH reactivity above the canopy:** From the 10[th] to the 13[th] and from the 15[th] to the 18[th] of July, above canopy measurements showed an average missing OH reactivity of 6.1 ± 5.0 s[-1] and a mean relative missing OH reactivity of 47.8%. While the day-time missing OH reactivity was on average 15.1 s[-1] for the "Wd" days, it was 5.5 s[-1] on average for the cooler days of "class Cd", keeping in mind that the day-time average might not be representative of the missing OH reactivity above canopy on warm days since it only includes one day of measurements. All nights corresponding to above canopy measurements were characterized by stable atmospheric conditions, which allows to see the effect of night-time temperature on the missing OH reactivity level, with the exception of the second part of the 17[th]-18[th] night, when the vertical turbulence increased leading to lower levels of OH reactivity and thus missing OH reactivity.





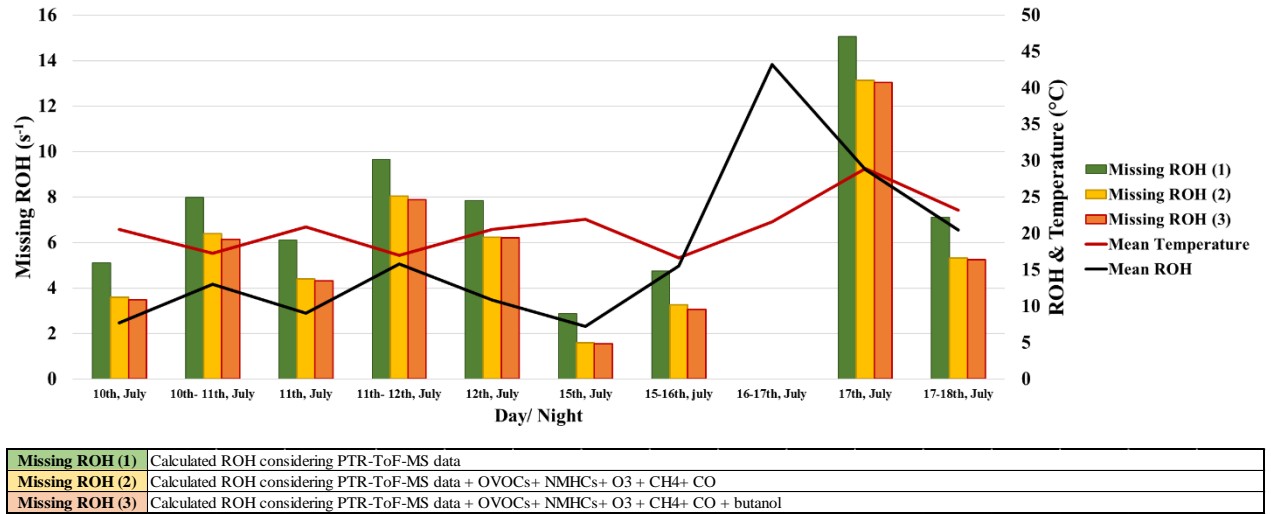

| Missing ROH (1) | Calculated ROH considering PTR-ToF-MS data |
| Missing ROH (2) | Calculated ROH considering PTR-ToF-MS data + OVOCs+ NMHCs+ O3 + CH4+ CO |
| Missing ROH (3) | Calculated ROH considering PTR-ToF-MS data + OVOCs+ NMHCs+ O3 + CH4+ CO + butanol |

**Figure 9: Evolution of missing OH reactivity above canopy with considered compounds in calculated OH reactivity.**

Considering other measurements performed at this height, the mean contribution of ozone, CO and methane was assumed to be 1.2 s$^{-1}$. NO$_x$ measurement was not performed at this height but their contribution at the 6 m height was 0.3 s$^{-1}$ on average, suggesting only a small contribution to the missing OH reactivity. Online chromatographic instruments (Table 2) provided information on other oxygenated (7 compounds) and non-methane hydrocarbons (36 compounds). These compounds could explain 0.45 s$^{-1}$ on average (0.34 s$^{-1}$ from NMHC and 0.11 s$^{-1}$ from OVOC measured by GC) of the missing OH reactivity between the 10$^{th}$ and the 12$^{th}$ of July. However, after the 14$^{th}$ of July, the GC measuring OVOC stopped working, but NMHCs alone covered 0.6 s$^{-1}$ on average. Thus, OVOC, NMHC, ozone, CO and methane could explain 1.7 s$^{-1}$ on average of the missing OH reactivity above the canopy. Finally, looking at butanol measured by the PTR-MS at the 12 m height, a maximum mean contribution of 0.3 s$^{-1}$ was assessed for the nights of 10$^{th}$-11$^{th}$ of July. Therefore, all the additional compounds mentioned above could represent between 7 and 24 % of the missing OH reactivity observed above the canopy. The contribution of the considered compounds is summarized in figure 9.

**Day-time missing OH reactivity:** To investigate the origin of the missing OH reactivity during the day, which could be due to primary emitted compounds or oxidation products of VOCs, we examined the variability of the missing OH reactivity, along with other compounds (acetic acid, sesquiterpenes, nopinone and pinonaldehyde as well as with the ratio of (MACR+MVK)/isoprene, knowing that MACR and MVK are oxidation products of isoprene.

First, for higher day-time missing OH reactivity observed for Wd days (within and above the canopy), Figure 7.c shows that the missing reactivity increases with acetic acid (max mixing ratio > 1.5 ppb). Acetic acid can be directly emitted by the trees and the soil (Kesselmeier and Staudt, 1999) and could also be an oxidation product of BVOCs, including isoprene (Paulot et al., 2011). This compound showed a diurnal cycle similar to that of isoprene (Fig 4.c), was not used to calculate the OH





reactivity. Despite its relatively low reactivity with OH, this compound showed a maximum calculated OH reactivity during Wd days (0.8 s$^{-1}$) that was, on average, 3.8 times higher than that of Cd days. Thus it could explain, with other compounds with similar behavior, part of the missing OH reactivity especially the ones seen during warm days. Secondly, the (MACR+ MVK)/ isoprene ratio showed a general trend with higher values during the day and lower values during the night, suggesting

that oxidation products of isoprene could be responsible of the day-time missing OH reactivity (S9). This ratio was generally higher for Wd days than for Cd reflecting a more intense photochemistry during warm days.

**Night-time missing OH reactivity:** The higher mixing ratios of acetic acid observed for Wd days, were also seen during S/Wn nights (4$^{th}$- 5$^{th}$ of July/ Inside canopy), as well as during the first hours of some SU/Wn nights (7$^{th}$- 8$^{th}$ of July/ Inside canopy,

17$^{th}$- 18$^{th}$ of July/ Inside and above canopy). During these days, the temperature started to decrease slowly around 17h local time and the night-time mean temperature was relatively high (mean T > 18°C). Following the temperature profile, isoprene and acetic acid mixing ratios decreased slowly and marked relatively high nocturnal levels. Thus, the significant missing OH reactivity observed during the mentioned nights could be related to unconsidered compounds characterized by a similar behavior than isoprene and acetic acid, which is clearly temperature dependent. It is worth noting that among warm nights, the

6$^{th}$-7$^{th}$ of July did not exhibit a significant missing OH reactivity. However, this night was characterized by a higher relative humidity (80-100%) than other nights, which may have led to a faster loss of OVOCs on humid surfaces. A close look at the MACR+ MVK profile in Figure 7c indicates that the level of isoprene oxidation products was lower than on other S/Wn and SU/Wn night, which is consistent with a higher loss rate on humid surfaces.

Furthermore, looking at the sesquiterpenes variability (Fig 7.e), they exhibited a similar trend as monoterpenes (Fig 4.c),

showing higher mixing ratios during night-time. These primary BVOCs were not considered in OH reactivity calculations but their maximum contribution to maximum night-time OH reactivity remained low, ranging between 0.2 and 1.2 s$^{-1}$ for stable nights. Interestingly, sesquiterpenes mixing ratios were higher inside the canopy compared to the above and the difference was significant during stable nights. Plotting the ratios sesqui(above)/MTs(above) with the ratios sesqui(inside)/MTs(inside) showed a slope of 0.72 and an R$^2$ of 0.5.  Knowing that sesquiterpenes are relatively highly reactive with ozone (Ciccioli et

al., 2002), which can dominate the chemistry during dark hours, this observation suggests that a fraction of these species could be consumed before reaching the above of the canopy, indicating a more intense night-time chemistry, leading to the formation of secondary compounds. This observation can be supported by monoterpenes oxidation products diurnal profiles (nopinone and pinonaldehyde), which exhibited higher values during stable/ warm nights and they could explain with other secondary products the missing OH reactivity seen during these nights (4$^{th}$-5$^{th}$ of July/Inside canopy). However as mentioned before, the

6$^{th}$-7$^{th}$ of July night was characterized by a high relative humidity which could explain the relatively lower levels of nopinone and pinonaldehyde and thus the non-significant missing OH reactivity, due to losses on humid surfaces. Finally, it is worth noting that Holzinger et al. (2005) reported the emission of highly reactive BVOCs in a coniferous forest, which is 6- 30 times the emission of monoterpenes on the studied site. This large fraction of BVOCs can be subject to oxidation by ozone leading



to unidentified, non-accounted for secondary molecules. These oxidation products can participate to the growth of new particles. Indeed, new particle formation episodes were recently reported on this site (Kammer et al., 2018).

To summarize, higher day-time missing OH reactivity was observed for warm days (Wd), inside and above the canopy, exhibiting a dependency on temperature profiles and showing that trace gases leading to the missing OH reactivity are linked

to an enhancement of primary species, which can also involve secondary products formation.  Regarding night-time missing OH reactivity, higher levels were seen for stable and warm nights showing that these conditions favor the accumulation of primary emissions as well as the reactions of ozone with biogenic species, leading to secondary products. Relative humidity, which was higher during dark hours, seemed to have an impact on missing OH reactivity due to a larger loss rate of oxygenated species on humid surfaces.

**4 Conclusion**

During summer 2017, total OH reactivity measurements were conducted as part of the LANDEX field campaign, in the Landes maritime pine forest (France). During this campaign, two instruments (LSCE-CRM and UL-FAGE) were deployed to measure total OH reactivity inside and above the canopy as well as at two different locations inside the canopy level. The comparison between both instruments, based on measurements done at the same location at the end of the campaign, showed

a good agreement (slope of 1.17 on a linear correlation plot). However, an offset of 4.2 s$^{-1}$ was obtained, which is potentially linked to an overestimation of the instrumental zero for the UL-FAGE instrument. Measuring at two different locations demonstrated a good horizontal homogeneity inside the canopy, even during episodes of vertical stratification that was observed during some nights.

Total OH reactivity recorded an average of 19.1 s$^{-1}$ at 6 m height, 1.2 times higher than that observed above the canopy level

at 12 m height. It varied similarly at both heights, following a diurnal cycle with two maxima, one during day-time following isoprene's profile and a higher one during night-time when monoterpenes concentrations reached their maxima. The later were the main emitted compounds in this forest ecosystem.

The variability of BVOC concentrations and OH reactivity were strongly dependant on meteorological parameters. Day-time OH reactivity was linked to ambient temperatures and light, two parameters governing the emissions of temperature and/ or

light dependent compounds (like isoprene), whereas night-time OH reactivity was influenced by night-time temperatures and vertical turbulence intensity. Indeed, low turbulence, high temperature and lower oxidation rates than during day-time, led to higher concentrations of monoterpenes and thus higher OH reactivity during stable and warm nights. In addition, higher differences in BVOCs levels and total OH reactivity were observed between the two studied heights particularly during stable nights.

Furthermore, monoterpenes showed to be the main contributors to total OH reactivity during both day-time and night-time. These species accounted for more than 60% of the OH reactivity during day-time, followed by isoprene (30%), acetaldehyde (3%) and MAC+ MVK (3%). However, the contributions of isoprene and OVOCs were much lower at both levels during the





night, leading to a higher contribution of monoterpenes, which was slightly more important inside the canopy level due to the stratified conditions.

An investigation of the missing OH reactivity indicated similar average levels at both heights over the whole campaign ($6.1 - 6.9$ $s^{-1}$). Higher missing OH reactivity was seen on warmer days inside and above the canopy, indicating that temperature-dependent primary emissions and/or the formation of unmeasured photo-oxidation products could be the cause of the missing OH reactivity. Regarding night-time missing OH reactivity, it was linked to vertical turbulence, ambient temperature and relative humidity. The highest night-time missing OH reactivity ($s^{-1}$) was observed during the stable/ warm night of the $4^{th}$- $5^{th}$ of July inside the canopy level. Complementary measurements performed inside ($O_3$, $NO_x$) and above the canopy (OVOCs, NMHCs, $O_3$, $NO_x$ and butanol), explained with methane and carbon monoxide, part of the missing OH reactivity, that remained significant for warm days and stable/ warm nights.

Similarly, sesquiterpenes and monoterpenes oxidation products (nopinone and pinonaldehyde) measured by PTR-MS accounted only for a small fraction of the observed missing OH reactivity during the night. However, lower mixing ratios of ozone during dark hours in the canopy could be explained by its consumption by the reaction with reactive BVOCs leading to the formation of unknown secondary products. This could be the case of sesquiterpenes that are highly reactive with ozone and which concentration above canopy is lower than the inside demonstrating a potential inter-canopy chemistry.

**Authors contribution:**

S. Bsaibes and F. Truong set up and carried out OH reactivity measurements with the LSCE-CRM. M. Al Ajami, C. Schoemaecker, S. Batut and C. Hecquet set up and carried out OH reactivity measurements with the UL-FAGE instrument. K. Mermet, T. Léornadis, S. Sauvage and N. Locoge carried out GC-BVOC2, GC-OVOCs and GC-NMHCs measurements and provided analyzed data. V. Gros provided GC-BVOC1 analyzed data. S. Dusanter carried out PTR-MS measurements and provided the corresponding data. J. Kammer and P.-M. Flaud provided NOx data and meteorological parameters. E. Villenave, E. Perraudin and P.-M. Flaud have coordinated the LANDEX project and field campaign. S. Bsaibes prepared the manuscript with the co-authors contribution, mainly M. Al Ajami, C. Schoemaecker, D. Dusanter, and V. Gros.

*Acknowledgements:* This study was supported by the European Union's Horizon 2020 research and innovation program under the Marie-Sklodowska-Curie grant agreement No 674911- IMPACT, ADEME- LANDEX and the CNRS. The authors want to acknowledge the Bilos ICOS team for meteorological data and site availability and Ineris for sharing their mobile laboratory. We would like to thank S. Schramm, D. Baisnée and R. Sarda-Esteve for their help during the installation of the PTR-Quad-MS and the GC-FID. The PC2A and SAGE participation was supported by the French ANR agency under contract no. ANR-11-LabX-0005-01 CaPPA (Chemical and Physical Properties of the Atmosphere), the Région Hauts-de-France, the Ministère de l'Enseignement Supérieur et de la Recherche (CPER Climibio) and the European Fund for Regional Economic Development.





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
