# Peer review of "Variability of OH reactivity in the Landes maritime Pine forest: Results from the LANDEX campaign 2017"

_Atmospheric Chemistry and Physics, 2019_

## Referee Comment (RC1) · Anonymous Referee #1 · 5 Jul 2019

This study shows very interesting results on reactivity and reactive compounds from pine forest in France. There is very nice measurement set-up and huge set of instruments to characterize reactive compounds in the forest air. Intercomparison of two totally different reactivity measurements showed good results and calculation of OH reactivity is well done and presented. Results are well presented and interpreted. I had mainly some concerns on the method for measuring mono- and sesquiterpenes and I hope some more information on the suitability of the GC method for these compounds can be provided. However, these were just ancillary measurements and I would recommend publishing with minor changes.

Specific comments:

Abstract lines 29-30: Could you also add a comment or a value how big fraction was missing?

Page 6, lines 15-20: How about O3? Did you apply any O3 correction? Have you detected any effect of O3 in your CRM system?

Page7, lines 27-28: Please, be more specific. What was the concentration range of isoprene and a-pinene?

Page 9, lines 28-29: Copper tubing impregnated with KI is commonly used for the DNPH measurements of aldehydes and ketones, but is it suitable for monoterpenes? Did you test the recovery of terpenes? What about particle filter? Do you see losses of terpenes in them? Maybe you could provide some reference on an earlier study where they have been tested.

Page 10, line 2 and 14: You used Carbotrap B and C for collecting terpenes. I am worried that they are not very good for mono- and sesquiterpenes and you may have some losses of them? Did you do some recovery tests? Have you detected any losses or isomerization while testing those? I would recommend for example Tenax TA cold trap for mono- and sesquiterpenes.

Page 10, line 12: In some of the MARKES Unity systems b-pinene and some other monoterpenes are isomerized and concentrations of some monoterpenes, for example p-cymene, are increasing over the time. Did you detect low response for b-pinene or for some other monoterpenes or increase of p-cymene?

Section 3.3.: Was the mean missing fraction higher inside or above canopy? I would guess there are more reaction products above the canopy.

Page 26, line 12: Is the typical B-value (0.057) for the monoterpene emissions or for the reactivity? Often B-value 0.09 is used for the monoterpene emissions.

Page 31, 14-15: I think that also for monoterpenes reactions with ozone can be very significant. Do you have any idea of OH radical concentrations at the site? It would be nice to know how much lower the lifetimes of VOCs were during the day and how important ozone reactions were. Sometimes ozone reactions can be very important also during the day.

Technical comments:

Table 1: Please, add an explanation to K' max

Page 10, line 13: You mention B-caryophyllene here, but it is not included into table 2 You have lots of time series plots, but they are a bit hard to follow and it would be also nice to get some quick and easy to look at average plots or tables (for example mean reactivity and mean missing reactivity during night and day, inside and above canopy and during cold and warm nights).

Page 28, line 16: ')' is missing

Page 28, line 21: Should this be 'This compound showed a diurnal cycle similar to that of isoprene (Fig 4.c) and was not used to calculate. . .'?

Page 29, line 5: What is '(S9)'?

Page 31, lines 8-10: I did not understand this sentence 'Complementary measurements performed inside (O3, NOx) and above the canopy (OVOCs,NMHCs, O3, NOx and butanol),explained with methane and carbon monoxide, part of the missing OH reactivity, that remained significant for warm days and stable/ warm nights.'

---

## Referee Comment (RC2) · Anonymous Referee #2 · 21 Jul 2019

The authors report OH reactivity measurements in a forested environment with measurements in and above the canopy. The contribution of measured and potentially unmeasured organic compounds is analysed. Results of this campaign are similar to previous observations at other locations and mainly confirm earlier conclusions. This is still interesting and within in the scope of ACP after addressing the following points: The readability of the manuscript could be improved. For example, for non-specialists of the measurement techniques, it will be hard to follow the description of instruments. Characterization experiments are not well described and results are not clear (for example, units are missing for slopes derived in linear regressions).

[Figure]

The characterization experiments for the CRM are described, but it remains unclear, how large corrections were. The authors should consider give some numbers, how big corrections were for typical chemical conditions of this campaign. A discussion about consequences for the accuracy of measurements would be beneficial.

The authors mention that one of the conclusions from previous campaigns were that potential loss of reactive VOCs could be a problem in CRM instruments. Did they quantitatively test this for example when they did the characterization experiment for the deviation from a pseudo-first order reaction system?

Similarly, did the authors test, if VOCs were quantitatively transmitted through inlet lines for the GC and PTR-MS analysis? How often were filters in inlet lines exchanged and did they authors test, if the transmission of VOC through filters decreased with time? The authors should mention early in the paper, how they deal with contributions of NO2 / NO to the OH reactivity.

Page 14 Point 3). It would be useful to give some numbers for the estimate of OH reactivity from species only measured at 12m height in the main text.

Figure 3: In a correlation plot, error bars of measurements are needed. Did the regression procedure take into account errors of the measurements?

P17 L19: How is the "higher vertical mixing leading to similar concentrations" quantified? The yellow frame (15 to 17 July) shows also large differences in monoterpene concentrations at different heights.

P17 L21: Which data are used for the linear regression discussed in this section? It does not sound likely that inhomogeneities of air masses result in a change in the intercept, but would increase the scatter of data in the correlation.

P18 L6: The reference Lou et al 2010 is not appropriate, because measurements in that paper were done in a mixed environment. P18 L20 / P20 L22: The authors may want to mention already here that it is well known that plant emissions are increasing

with increasing temperature.

Section 3.3/3.5.: The discussion would benefit, if the accuracy of calculated OH reactivity were taken into account (maybe also shown in Fig. 5). Is there an estimate of OH reactivity from oxidation products not taken into account here (for example from oxidation products like MVK/MACR)? Is there any estimate, if transportation from other sources could have been impacted the location?

Section 3.4.: Would the authors expect a difference in the distribution of OH reactants? Was there any attempt to estimate how much of the emissions were oxidized inside the canopy?

P29: Sesquiterpene oxidation products are likely not measured. Could the authors still estimate how much reactivity would be expected, if the difference between in and above canopy was due to oxidation?

Figures in the main text and supplementary material: Font sizes are very small. It would be easier for the reader, if they were larger. The position of legend below the x-axis label is unusual. Errors bars of measurements would be helpful to judge differences, if quantities are compared.

Technical: The authors should follow the style of the journal for example how figures are referenced, dates are given and SI units should be used.
* * *

---

## Referee Comment (RC3) · Anonymous Referee #3 · 29 Jul 2019

Bsaides et al. presented ambient observed OH reactivity datasets from two instrumentation – CRM and LP LIF. The field site is a coniferous temperate forest site in south-western France. They presented intercomaprison results between CRM and LP-PIF followed by a series of data analysis for the observed differences between the inside and the outside of the canopy in addition to between the day and the night. They particularly utilized a turbulence index as a determinant for a stagnation causing higher total observed OH reactivity likely due to the suppressed vertical mixing. It is not a trivial work to have all the listed instrumentation in one field site to get a comprehensive dataset as presented. However, the presentation and the analysis of the dataset can be improved substantially to provide better scientific insights. The list for the suggested

improvements are following.

1) The conclusion of this study is obviously hand waving as they conclude that the origin of mixing OH reactivity is either uncharacterized emission or oxidation products. Those are basically the nature of all VOCs in the atmosphere anyway. A deeper discussion may be utilizing a box model is recommended to narrow down the source of missing OH reactivity

2) It is not entirely clear whether ambient VOC samples and OH reactivity samples were collected with the same sampling tubes. Please clarify this point as it is very important to evaluate potential imparity

3) As the oxidation product of CO is $HO_2$, it is more likely susceptible to interference from OH recycling during the calibration process with high CO concentrations. What CO levels do you use for calibration? Could you provide at least simple discussion that was not the case in your calibration process?

4) It appears that the trace gas OH reactivity such as CO, NOX, O3 and SO2 is not considered in the calculated OH reactivity assessments. Considering the rural location, this may not be a substantial factor, but it still requires to be included.

5) Page 13 Line 12: Further quantitative discussion on the impacts from MT to the isoprene mass. What species would be susceptible for the fragment and how prevalent it can be?

6) It is well known that PTR sees higher MT then the sum of speciated MT quantified by GC. Add this discussion whether that was the case during the observational period. This may give us some insight on the missing OH reactivity.

7) Page 14 Line 3: Further quantitative discussion is required. It is not clear how the 4 % value has been drawn.

8) Figure 2: it is extremely confusing what I should look up to for the comparison. It would be better separate into figures describing in the different periods. I would

recommend to present an intercomparison figure first so that readers can get a sense on the potential bias from the instrumentation. Also, please make it clear which MT species are consisting the total MT presented in the figure.

9) Figure 3: If you take a diurnal average and adjust the intercept, then do two diurnal variations agree better? It seems CRM has 4 s-1 offset but the text description says otherwise. Please make them consistent! In addition, even without the intercept, there are $\sim$ 20 % differences in the relationship. Please discuss the potential reasons!

10) A more description on u*is required: how you measured them and justify the classifications.

11) Page 20 line 12: Have you seen the described extreme weather events during the observations? If you have not, then this discussion is irrelevant.

12) The stable nocturnal boundary layer could cause accumulation of long-lived oxidation products of VOCs instead of vertical mixing. Therefore, the speculation for the MT emission attributing missing OH reactivity is not conclusive. The authors need to substantiate argument.

The dataset worth thorough investigation for better understanding in potential roles of MT in atmospheric reactivity. I would recommend the authors to take more time for the major revision of the manuscript.

---

## Author Comment (AC1) · 5 Oct 2019

1- Abstract lines 29-30: Could you also add a comment or a value how big fraction was missing?

New version page 1, lines 31- 32: Comparing the measured and the calculated OH reactivity highlighted an average missing OH reactivity of 22% and 33%, inside and above the canopy, respectively.

2-Page 6, lines 15-20: How about O3? Did you apply any O3 correction? Have you detected any effect of O3 in your CRM system?

- Based on previous experiments (Fuchs et al., 2017), no ozone dependency was seen for the LSCE-CRM. Therefore, no tests were performed to characterize the interference due to O3 and no correction was applied to OH reactivity raw data. This information has been added in the revised version of the manuscript as:

New version page 6, lines 176- 179: In some CRM systems, corrections for potential NO2 and/or O3 artefacts are also considered (Michoud et al., 2015, Praplan et al., 2017). On one hand, NO2 is subject to photolysis leading to NO, which can subsequently react with HO2 yielding OH. On the other hand, O3 can also be photolyzed in the reactor, producing O(1D), which reacts further with H2O, yielding two OH radicals.

And page 8, lines 228 -232: NO mixing ratios were lower than 0.5 ppb (corresponding to the detection limit of the NOx monitor deployed during LANDEX) most of the time for the measurement time periods used in this study, and no correction was applied for the spurious formation of OH from the HO2+NO reaction. Similarly, for NO2, no correction was applied due to the low ambient mixing ratio of $1.1 \pm 0.8$ ppb. Regarding O3, no dependency was seen for LSCE-CRM, based on previous experiments (Fuchs et al., 2017). Therefore, no correction was applied. The correction (D) on the reactivity values due to the dilution was around 1.46 during the campaign.

3-Page7, lines 27-28: Please, be more specific. What was the concentration range of isoprene and a-pinene?

New version, page 7 lines 221 - 223 : To determine the correction factor for the deviation from pseudo-first order kinetics, injections of known concentrations of isoprene (k isoprene+OH = 1x10-10 cm3 molecule-1 s-1, 1- 120 ppb) and $\alpha$-pinene (k $\alpha$-pinene+OH = 5.33 x 10-11 cm3 molecule-1 s-1, 3 -190 ppb) (Atkinson, 1985) were performed before and after the field campaign since they represent the dominant species in this forest ecosystem.

4-Page 9, lines 28-29: (a) Copper tubing impregnated with KI is commonly used for the DNPH measurements of aldehydes and ketones, but is it suitable for monoterpenes?

Did you test the recovery of terpenes?

- As presented in Mermet et al. 2019 (AMTD), several tests were performed on scrubbers recommended by ACTRIS (copper tubes coated with potassium iodide, glass filters impregnated with sodium thiosulfate, and copper screens coated with manganese dioxide) to characterize (1) O3 removal efficiency, (2) losses of BVOCs in the absence of ozone, and (3) potential ozone-induced losses of BVOCs in the scrubber. Copper tubes coated with potassium iodide (KI) appeared as the best choice for BVOC measurements. In the absence of ozone, KI scrubbers exhibited BVOC losses lower than 5% for most non-oxygenated species, whereas in the presence of ozone, losses were relatively higher but remained lower than 15% (lower than 5% for $\alpha$- and $\beta$-pinene). The only two notable exceptions were the most reactive compounds, i.e. $\alpha$-terpinene and $\beta$-caryophyllene, whose losses were approximately 20% and 40%, respectively. These two species represent only a minor fraction (3% maximum) of the total sum of compounds measured with GC-BVOC2 inside the canopy, compared to maxima of 42-43% for $\alpha$ and $\beta$-pinene.

(b) What about particle filter? Do you see losses of terpenes in them?

- No tests were made on the particle filters. ACTRIS 2014 measurement guidelines were followed. High flow rates were set in the sampling lines: 1 L min-1 for GC instruments and 10 L min-1 for the PTR-MS. The contact time between ambient BVOCs and the particle filters is extremely short and we don't expect significant losses.

(c) Maybe you could provide some reference on an earlier study where they have been tested. - ACTRIS. 2014. "WP4- NA4: Trace Gases Networking: Volatile Organic Carbon and Nitrogen Oxides Deliverable D4.9: Final SOPs for VOCs Measurements." ACTRIS.

This information has been added in the revised version of the manuscript as:

New version Page 10, lines 304- 311: Measurements of VOCs (Table 3) were performed at different locations (Figure 1) by a proton transfer reaction-mass spectrometer (PTR-MS) and four on-line gas chromatographic (GC) instruments. Ozone scrubbers (Copper tube impregnated with KI) and particle filters were added to the inlets of all GC sampling lines. Losses of BVOCs in these ozone scrubbers were investigated under similar sampling conditions in the absence and presence of O3 (Mermet et al., 2019, AMTD). The scrubbers exhibited less than 5% losses for most non-oxygenated BVOCs, whereas in the presence of ozone, losses were relatively higher for some BVOCs, but remained lower than 15% (lower than 5% for $\alpha$- and $\beta$-pinene). High flow rates were applied in the sampling lines: 1 L min-1 for GC instruments and 10 L min-1 for the PTR-MS, therefore, the contact time between ambient BVOCs and the particle filters was extremely short and no significant losses are expected.

5- Page 10, line 2 and 14: You used Carbotrap B and C for collecting terpenes. I am worried that they are not very good for mono- and sesquiterpenes and you may have some losses of them? Did you do some recovery tests? Have you detected any losses or isomerization while testing those? I would recommend for example Tenax TA cold trap for mono- and sesquiterpenes.

- Carbotrap C in GC-BVOC1 is already set by the manufacturer. Carbotrap B has been selected among the possible adsorbent as listed in the ACTRIS guidelines (ACTRIS, 2014). The method has been optimized in terms of temperature of the thermodesorption, the column, the sampling volume and sampling line including a scrubber. Results are shown in the Mermet et al. AMTD, 2019. Based on a reference mixture composed of 14 monoterpenes, tests resulted in a good separation for most of the compounds. Apart sabinene and terpinene, a good recovery has been obtained between the experimental response coefficient compare to the theoretical ones (determined from the Equivalent carbon number for FID). As a consequence, the calculated uncertainties are significantly higher for these 2 compounds, for which some isomerization or thermodegradation could occur. Indeed, Tenax TA is another well characterized adsorbent but thermodegradtion of monoteprenes may also occur as reported by Coeur et al. (1997).

This information has been added in the revised version of the manuscript as:

New version page 11, lines 333- 335: The method has been optimized in terms of temperature of the thermodesorption, the column, the sampling volume and sampling line including a scrubber. More details about the optimization and the tests performed can be found in Mermet et al. AMTD, 2019.

6- Page 10, line 12: In some of the MARKES Unity systems b-pinene and some other monoterpenes are isomerized and concentrations of some monoterpenes, for example p-cymene, are increasing over the time. Did you detect low response for b-pinene or for some other monoterpenes or increase of p-cymene?

- p-cymene response observed was elevated comparing to other monoterpenes. For some monoterpenes a low response was observed. It is the case of sabinene, ter-pinolene, 2-carene for example, but not for the most abundant monoterpenes such as b-pinene, a-pinene, limonene or myrcene (Mermet et al., 2019). While isomerization may be an issue for measuring some monoterpenes with this instrument, the most abundant contributors to the OH reactivity are well measured and this issue does not impact the conclusions of this study. The method could be optimized by using another desorption system. To take into account, the question of the reviewer, in the revised manuscript we refer the reader to the paper of Mermet et al. which gives all the results concerning the optimization and the tests which have been performed.

This has been added to the new version of the paper page 11, lines 334- 335: More details about the optimization and the tests performed can be found in Mermet et al. AMTD, 2019.

7- Section 3.3.: Was the mean missing fraction higher inside or above canopy? I would guess there are more reaction products above the canopy.

- Section 3.3 aims to present a comparison between measured and calculated OH reactivity whereas missing reactivity (as absolute and relative fractions) is discussed in

section 3.5. The mean relative missing fraction was around 48% above the canopy and 38% inside the canopy, when comparing the measured OH reactivity with the calculated one from PTR-MS data, which was measuring at both heights However, it should be reminded that, measurements were not performed simultaneously above and inside the canopy, except for a short period from mid-day of the 17th, July to mid-day of the 18th, July.

This information is mentioned in the text:

New version page 28, lines 748- 749: When comparing measurements of OH reactivity with calculations based on PTR-MS data (see Table 3), an average of 38% (7.3 s-1) and 48%. (6.0 s-1), remained unexplained inside and above the canopy, respectively.

8- Page 26, line 12: Is the typical B-value (0.057) for the monoterpene emissions or for the reactivity? Often B-value 0.09 is used for the monoterpene emissions.

The $\beta$ value is normally used for monoterpenes emissions from vegetation. When applied on missing OH reactivity data, it can be used to indicate if the missing OH reactivity is linked to primary emissions that are temperature-dependent like monoter-penes. When the measured ROH was compared to the calculated one from PTR-MS data, a $\beta$ of 0.09 was obtained when the missing ROH was fitted in the equation used to describe the temperature dependency of monoterpenes emissions. This $\beta$ was in the range of $\beta$-values normally seen for monoterpenes emissions. However, following the remark of reviewer 3, we have decided to examine the missing reactivity by taking into account in the calculated reactivity all the measured compounds available at the 6 m height. In this case, the missing was also fitted in the exponential relation, but the $\beta$ value was higher (0.17), which indicates that the missing is not only linked to primary emissions but is also due to secondary oxidation products (Mao et al., 2012, Hansen et al. 2014, Kaiser at al.,2016).

New version page 29, lines 775- 789: As reported in Di Carlo et al. (2004), the missing OH reactivity was fitted with an equation usually used to describe temperaturedependent emissions of monoterpenes (Guenther et al., 1993): E(T) = E (293) exp($\beta$(T-293)), where E(T) and E(293) represent the emission rate at a given temperature T and at 293K, respectively. In this equation, E(T) was substituted to MROH(T) and E (293) by MROH (293) with MROH representing the missing OH reactivity (Hansen et al., 2014). The value of $\beta$ determined from the fit of the data for the 6 m height (day-time), is around 0.17, higher than the values attributed to monoterpenes emissions from vegetation (0.057 to 0.144 K-1). Higher $\beta$-values were also obtained by Mao et al. (2012), Hansen et al. (2014) and Kaiser et al. (2016), were they suggested that day-time missing reactivity is mostly linked to secondary oxidation products. However, the use of $\beta$ factor must be made with caution, as the missing OH reactivity can be influenced by processes that do not affect BVOCs emissions (i.e. the boundary layer height and the vertical mixing). Furthermore, we cannot exclude the possibility of light and temperature dependent emissions. Indeed, Kaiser et al. (2016) also investigated the temperature dependency of day-time missing OH reactivity in an isoprene-dominated forest, reporting that part of the missing emissions could be characterized by a light and temperature dependence, knowing that temperature increases with increasing solar radiation. Regarding above canopy, most measurements were performed during cool days. Thus, it was not possible to analyze the temperature dependence of above canopy day-time missing OH reactivity.

9- Page 31, 14-15: I think that also for monoterpenes reactions with ozone can be very significant. Do you have any idea of OH radical concentrations at the site? It would be nice to know how much lower the lifetimes of VOCs were during the day and how important ozone reactions were. Sometimes ozone reactions can be very important also during the day.

- Based on the referee's comment, calculations of $\alpha$-pinene lifetime (one of the major compounds) towards OH and O3 were made.

Information has been added in the new version of the paper, page 26, lines 723- 734 :

The concentration of OH was 4.2×106 molecules cm-3 on average during day-time with a maximum of 4.3×107 molecules cm-3 and around 1.5×106 molecules cm-3 on average during night-time (data available between the 13th and the 19th, July). However, a potential artefact on OH radical's measurements leading to a possible overestimation of OH radical's concentrations, could not be ruled out. Regarding ozone, its mixing ratio showed a diurnal cycle with maximum values during the day (max ≈ 60 ppbv, mean ≈ 29 ppbv), that were similar within and above the canopy due to efficient mixing, and lower levels during nights, with an average of 18 ppbv inside canopy, while levels higher by 1 - 9 ppb on average, above the canopy. Considering OH and O3 average mixing ratios, the $\alpha$-pinene lifetime was estimated to be 1.2 hours and 4 hours, respectively, during the day, and 3.6 hours and 5.8 hours, respectively, during the night. At maximum OH and O3 mixing ratios during day-time, the $\alpha$-pinene lifetime was reduced to 7.4 min and 2 hours, respectively. Thus, OH chemistry remained dominant compared to ozonolysis of main emitted compounds on this site (i.e. $\alpha$-pinene). An article on the reactivity of monoterpenes with OH, ozone and nitrate for this campaign is in preparation (Mermet et al., in preparation).

Technical comments:

10- Table 1: Please, add an explanation to K' max

New version (Table 1): ROH max (s-1) instead of K' max (s-1).

11-Page 10, line 13: You mention B-caryophyllene here, but it is not included into table 2 It should be removed from the text.

- B-caryophyllene was added in Table 3 of the revised paper.

New version: Page 11, lines 325- 326: It was used to monitor 20 C5-C15 BVOCs, including isoprene, $\alpha$- and $\beta$-pinene, carenes and $\alpha$-phellandrene at the 6 m height with a time resolution of 90 min.

12-You have lots of time series plots, but they are a bit hard to follow and it would be

also nice to get some quick and easy to look at average plots or tables (for example mean reactivity and mean missing reactivity during night and day, inside and above canopy and during cold and warm nights).

- A table has been added in the new version of the paper: Page 28: Table 4. Summary of the measured OH reactivity and the missing OH reactivity inside and above the canopy, during the day and the night, taking into account only PTR-MS data or all the data available at each height for OH reactivity calculations. These averages are calculated for the periods when CRM, PTR-MS and others instruments data are available.

- A more detailed table has been added in the supplementary material: Table S9

13- Page 28, line 16: ')' is missing. - Corrected.

14- Page 28, line 21: Should this be 'This compound showed a diurnal cycle similar to that of isoprene (Fig 4.c) and was not used to calculate . . .'?

Indeed. New version, page 30, lines 810- 811: This compound showed a diurnal cycle similar to that of isoprene (Fig 4.c) and was not used to calculate the OH reactivity.

15- Page 29, line 5: What is '(S9)'? S9 is supplementary material 9.

16- Page 31, lines 8-10: I did not understand this sentence 'Complementary measurements performed inside (O3, NOx) and above the canopy (OVOCs,NMHCs, O3, NOx and butanol),explained with methane and carbon monoxide, part of the missing OH reactivity, that remained significant for warm days and stable/ warm nights.'

This part of the conclusion was modified:

13- Page 28, line 16: ')' is missing. Corrected.

14- Page 28, line 21: Should this be 'This compound showed a diurnal cycle similar to that of isoprene (Fig 4.c) and was not used to calculate . . .'? Indeed. New version: This compound showed a diurnal cycle similar to that of isoprene (Fig 4.c) and was not

used to calculate the OH reactivity.

15- Page 29, line 5: What is '(S9)'? S9 is supplementary material 9.

16- Page 31, lines 8-10: I did not understand this sentence 'Complementary measurements performed inside (O3, NOx) and above the canopy (OVOCs,NMHCs, O3, NOx and butanol),explained with methane and carbon monoxide, part of the missing OH reactivity, that remained significant for warm days and stable/ warm nights.'

This part of the conclusion was modified:

An investigation of the missing OH reactivity indicated averages of 6.0 and 7.3 s-1 inside and above the canopy, respectively, over the whole campaign. However, it showed some diurnal variability at both heights. During day-time, higher missing OH reactivity was observed on warmer days inside and above the canopy. Plotted against temperature, inside canopy missing OH reactivity showed a dependency on temperature. The analysis suggested that the missing OH reactivity may be due to unmeasured primary emitted compounds and oxidation products. In this context, OH reactivity measurements from a Pinus pinaster Aiton branch enclosure, could be of great interest to verify the contribution of unaccounted/unmeasured BVOCs emissions to OH reactivity as done by Kim et al. (2011), for red oak and white pine branch enclosures. Furthermore, higher levels of isoprene oxidation products on warmer days also suggest that the missing reactivity could be due to the formation of unmeasured oxidation products. Regarding the night-time period, the highest missing OH reactivity was found inside canopy for the 4th-5th, July night. This night was characterized by higher levels of isoprene and its oxidation products, compared to the night of the 6th-7th, July with similar atmospheric conditions. Air masses backward trajectories showed a continental origin for this night, suggesting that species, emitted by the largely spread Landes forest, could have been imported to the site and accumulated due to the stable nocturnal boundary layer. These species, unmeasured by the deployed analytical instruments and hence not considered in OH reactivity calculations, could explain the higher

missing OH fraction for the 4th-5th, July night. Finally, the investigation of sesquiter-penes and monoterpenes oxidation products (nopinone and pinonaldehyde) measured by PTR-MS highlighted their small contribution in terms of OH reactivity. They only explained a small fraction of the observed missing OH reactivity inside and above canopy during night.

References: - Mermet, K., Sauvage, S., Dusanter, S., Salameh, T., Léonardis, T., Flaud, P.-M., Perraudin, É., Villenave, É., and Locoge, N.: Optimization of a gas chromatographic unit for measuring BVOCs in ambient air, Atmos. Meas. Tech. Discuss., https://doi.org/10.5194/amt-2019-224, in review, 2019 - ACTRIS, 2014. WP4-NA4: Trace gases networking: Volatile organic carbon and nitrogen oxides Deliverable D4.9: Final SOPs for VOCs measurements. ACTRIS. - Coeur, C., Jacob, V., Denis, I., Foster, P., 1997. Decomposition of $\alpha$-pinene and sabinene on solid sorbents, tenax TA and carboxen. J. Chromatogr. A 786, 185–187. https://doi.org/10.1016/S0021-9673(97)00562-1 - Atmospheric Reactivity of Biogenic Volatile Organic Compounds in a Maritime Pine Forest during the LANDEX Field Campaign Kenneth Mermet, Emilie Perraudin, Sébastien Dusanter, Stéphane Sauvage, Thierry Léornadis, Pierre-Marie Flaud, Sandy Bsaibes, Julien Kammer, Vincent Michoud, Aline Gratien, Manuela Cirtog, Mohamad Al Ajami, François Truong, Sébastien Batut, Christophe Hecquet, Jean-Francois Doussin, Coralie Schoemaecker, Valérie Gros, , Nadine Locoge and Eric Villenave, in preparation.

Please also note the supplement to this comment:
https://www.atmos-chem-phys-discuss.net/acp-2019-548/acp-2019-548-AC1-supplement.pdf

[Figure]

**Supplement:**

**Table 4.** Summary of the measured OH reactivity and the missing OH reactivity inside and above the canopy, during the day and the night, taking into account only PTR-MS data or all the data available at each height for OH reactivity calculations. These averages are calculated for the periods when CRM, PTR-MS and others instruments data are available.

| | Mean Measured OH reactivity (s$^{-1}$) | Mean missing OH reactivity with PTRQi-ToFMS (s$^{-1}$) | Missing ROH considering PTRQi-ToFMS data + other measurements (s$^{-1}$) |
|---|---|---|---|
| **Inside** | 19.0 | 7.3 | 4.3 |
| Day | 18.4 | 7.0 | 4.1 |
| Night | 21.4 | 9.0 | 5.6 |
| Stable cool nights | 20.5 | 5.7 | 2.1 |
| Stable warm nights | 41.6 | 10.9 | 6.9 |
| Unstable cool nights | 7.9 | 4.5 | <LOD |
| Unstable warm nights | 13.5 | 6.8 | 3.6 |
| **Above** | 12.6 | 6.0 | 4.2 |
| Day | 10.4 | 5.0 | 3.1 |
| Night | 15.5 | 7.5 | 5.6 |
| Stable cool nights | 14.8 | 7.5 | 5.7 |
| Stable warm nights | ____ | ____ | ____ |
| Unstable warm nights | 20.5 | 7.1 | 5.2 |
| Unstable cool nights | ____ | ____ | ____ |

**Table S9: Summary of day/ night mean values of measured OH reactivity, the calculated one from PTR-MS data and from all available measurements, as well as the resulting missing OH reactivity, inside and above the canopy.**

| | Day/ Night | Measured ROH/ LSCE- CRM (s$^{-1}$) | Calculated ROH PTRQi-ToFMS (s$^{-1}$) | Calculated ROH PTRQi-ToFMS + other measurements (s$^{-1}$) | Missing ROH considering PTRQi-ToFMS (s$^{-1}$) | Missing ROH considering PTRQi-ToFMS + other measurements (s$^{-1}$) | Mean Temperature (°C) | Mean u* (m/s) | Day/ Night state |
|---|---|---|---|---|---|---|---|---|---|
| Inside canopy | 3rd, July | 14.5 | 5.3 | 7.8 | 9.2 | 6.8 | 22.9 | 0.4 | Cool |
| | 3rd- 4th, July | 25.7 | 16.1 | 20.1 | 9.6 | 5.6 | 15.6 | 0.1 | Stable/ Cool |
| | 4th, July | 20.0 | 9.9 | 12.2 | 10.1 | 7.7 | 26.5 | 0.6 | Warm |
| | 4th- 5th, July | 45.7 | 28.2 | 32.6 | 17.5 | 13.1 | 21.7 | 0.2 | Stable/ Warm |
| | 5th, July | 16.7 | 8.1 | 10.8 | 8.7 | 6.1 | 28.1 | 0.4 | Warm |
| | 5th- 6th, July | 12.6 | 4.8 | 8.4 | 7.8 | 4.2 | 18.9 | 0.3 | Unstable/ Stable/ Warm |
| | 6th, July | 22.1 | 11.0 | 13.9 | 11.1 | 8.2 | 24.2 | 0.4 | Warm |
| | 6th- 7th, July | 37.5 | 33.2 | 36.9 | 4.4 | < LOD | 20.1 | 0.1 | Stable/ Warm |
| | 7th, July | 28.1 | 18.5 | 21.6 | 10.0 | 6.5 | 28.3 | 0.4 | Warm |
| | 7th- 8th, July | 17.9 | 9.6 | 13.1 | 8.7 | 4.8 | 21.2 | 0.4 | Unstable/ Warm |
| | 8th, July | 13.2 | 7.5 | 10.3 | 5.7 | 2.9 | 23.0 | 0.5 | Cool |
| | 8th- 9th, July | 7.3 | 2.5 | 5.3 | 4.7 | < LOD | 19.5 | 0.5 | Unstable/ Warm |
| | 9th, July | 7.4 | 3.0 | 5.3 | 4.4 | < LOD | 20.9 | 0.8 | Cool |
| | 9th -10th, July | 7.9 | 3.4 | 7.1 | 4.5 | < LOD | 18.2 | 0.3 | Unstable/ Stable/ Cool |
| Above canopy | 10th, July | 7.7 | 2.6 | 4.3 | 5.1 | 3.4 | 20.6 | 0.5 | Cool |
| | 10th-11th,July | 13.0 | 5.1 | 6.9 | 8.0 | 6.1 | 17.3 | 0.1 | Stable/ Cool |
| | 11th, July | 9.5 | 3.4 | 5.2 | 6.1 | 4.3 | 20.9 | 0.4 | Cool |
| | 11th- 12th, July | 15.8 | 6.1 | 7.9 | 9.7 | 7.9 | 17.0 | 0.1 | Stable/ Cool |
| | 12th, July | 10.9 | 3.0 | 4.7 | 7.9 | 6.2 | 20.5 | 0.7 | Cool |
| | 12th- 13th, July | | | | | | 18.0 | 0.1 | Stable/ Cool |
| Inside canopy | 13th July | 6.7 | 3.1 | 6.3 | 3.6 | <LOD | 20.1 | 0.4 | Cool |
| | 13th- 14th, July | 18.7 | 15.3 | 19.1 | 3.4 | <LOD | 18.0 | 0.1 | Stable/ Cool |
| | 14th, July | 8.9 | 6.7 | 9.1 | < LOD | <LOD | 21.2 | 0.5 | Cool |
| | 14th- 15th, July | 17.1 | 12.9 | 16.0 | 4.2 | <LOD | 15.2 | 0.1 | Stable/ Cool |
| | 15th, July a.m | 13.3 | 12.6 | 15.6 | < LOD | <LOD | 22.0 | 0.4 | Cool |
| Above canopy | 15th, July p.m | 7.0 | 4.2 | 5.9 | 2.9 | <LOD | | | |
| | 15th- 16th, July | 15.6 | 10.8 | 12.7 | 4.8 | 2.9 | 16.6 | 0.1 | Stable/ Cool |
| | 16th, July | 10.0 | 8.1 | 9.9 | <LOD | <LOD | 26.7 | 0.5 | Warm |
| | 16th- 17th, July | | | | | | 21.6 | 0.1 | Stable/ Warm |
| 1h Inside/ 1h Above canopy | 17th, July | 41.5/ 39.8* | 23.3/ 23.0* | 26.8/ 25.0* | 18.2/ 16.8* | 14.7/ 14.8* | 28.9 | 0.4 | Warm |
| | 17th-18th, July | 20.5/ 20.5* | 15.3/ 13.4* | 18.4/ 15.3* | 5.2/ 7.2* | ≈ LOD/ 5.2* | 23.2 | 0.3 | Unstable/ Stable/ Warm |
| | 18th, July | 11.5/ 8.4* | 7.9/ 6.4* | 10.3/ 8.1* | 3.6/ < LOD* | <LOD/ < LOD* | 30.3 | 0.5 | Warm |

---

## Author Comment (AC2) · 5 Oct 2019

We thank referee (2) for the valuable comments.

1- The characterization experiments for the CRM are described, but it remains unclear, how large corrections were. The authors should consider give some numbers, how big corrections were for typical chemical conditions of this campaign. A discussion about consequences for the accuracy of measurements would be beneficial.

New version, page 8 (Table in the attached file), lines 237 - 240: Table 2 reports a summary of the corrections resulting from our tests and their impact on measurements. As

shown in table 2, the application of (F), for the deviation from pseudo-first order kinetics, induces the largest correction, with an absolute increase of 10.02 s-1 on average. Furthermore, this factor (F) has the largest relative uncertainty, with ±36%, against ±2% for the humidity correction factor.

2-The authors mention that one of the conclusions from previous campaigns were that potential loss of reactive VOCs could be a problem in CRM instruments. Did they quantitatively test this for example when they did the characterization experiment for the deviation from a pseudo-first order reaction system?

- In order to minimize potential losses of reactive VOCs in the CRM sampling system, heated (≈50°C) sulfinert lines were used. Indeed, Kim et al. (2009), showed that losses of b-caryophyllene are negligible in heated lines with temperatures above 20°C. More details are also mentioned in the answer to comment 3.

Information has been added in the revised version of the manuscript as:

Page 7, line 199- 201: Ambient air was sampled through two 1/8" OD sulfinert lines, collocated on a mast close to the trailer (see Fig. 1(a)). The lines lengths were 8 m for the measurements performed inside the canopy and 12 m for those performed above. These lines were heated up to 50°C as it was shown that losses of highly reactive molecules (b-caryophyllene) were negligible for temperature above 20°C (Kim et al., 2009).

3- Similarly, did the authors test, if VOCs were quantitatively transmitted through inlet lines for the GC and PTR-MS analysis? How often were filters in inlet lines exchanged and did they authors test, if the transmission of VOC through filters decreased with time?

- For GC instruments, VOCs were sampled through sulfinert sampling lines, similar to those used in the CRM sampling system, heated up to 50°C, with a flow rate of, at least, 1 L min-1, ensuring a short residence time of less than 8s. Materials used

are recommended by ACTRIS guidelines (ACTRIS 2014). Regarding the PTR-MS, the sampling lines were made of PFA (1/4"-OD) and were heated at 50°C. All lines were 15-m long and the flow rates were adjusted to 10 L min-1 to reduce the residence time below 2-s. Filters were also made of PFA and were changed every 2-weeks. No tests were performed to check the transmission of VOCs. However, Kim et al. (2009) tested losses of $\beta$-caryophyllene (a sticky sesquiterpene) in a 40-m long Teflon tube (1/4"-OD) flushed at 25 L min-1. These operating conditions lead to a residence time similar to that observed during LANDEX for our PTR-MS sampling system. The authors varied the line temperature from zero to 40°C using a temperature controlled environmental chamber and showed that losses of $\beta$-caryophyllene are negligible above 20°C. The PTR-MS lines being heated to 50°C in this study, no losses are expected for VOCs reported in this study.

This information has been added in the revised version of the manuscript as:

Page 12, lines 356- 359: Sulfinert material chosen for all GCs sampling lines and used in LSCE-CRM sampling system, is recommended by ACTRIS 2014. High flows were set in the lines (residence time of less than 8 s), that were heated up to 50°C to minimize the losses of potential reactive species. Filters and scrubbers were changed twice for the GC-BVOC1 and one time for the other GC instruments.

Page 12, lines 364- 366: The lines (PFA, 1/4" OD) were heated at 50°C and constantly flushed at 10 L min-1 using an additional pump and rotameters. Indeed, Kim et al. (2009) tested losses of $\beta$-caryophyllene in similar operating conditions. Authors varied the temperature from zero to 40°C showing that losses of b-caryophyllene are negligible above 20°C. The residence time was lower than 2s.

4- The authors should mention early in the paper, how they deal with contributions of NO2 / NO to the OH reactivity.

New version page 16, lines 469- 474:

A large range of NMHCs and OVOCs were measured at the 12 m height only by GC-NMHC and GC-OVOC (Table 3). Butanol (from SMPS exhausts) was also checked and found to be negligible at 12 m and highly and rapidly variable at 6 m (short peaks). NO and NO2 were only measured at the 6 m height. Mean NO mixing ratio was below the LOD for the measurement period and NO2 was around 1.1 ± 0.8 ppb on average. Thus, it was chosen not to take these species into account in the OH reactivity calculations, since they are not available at both levels. However, sensitivity tests were performed, in order to compute the relative contribution of butanol, OVOCs and NMHCs to OH reactivity (See section 3.5 and Fig. S5 and S6).

5- Page 14 Point 3). It would be useful to give some numbers for the estimate of OH reactivity from species only measured at 12m height in the main text.

- This paragraph (point 3, page 16 of the revised paper) describes the methodology used. No results were included. The contribution of species only measured at 12m to OH reactivity is mentioned on page 28 of the revised version, when investigating the missing OH reactivity.

6- Figure 3: In a correlation plot, error bars of measurements are needed. Did the regression procedure take into account errors of the measurements?

Errors bars were added as shown in Fig. 3, page 18 of the revised version (in attachment). Errors of the measurements were not taken into account in the regression procedure.

7- P17 L19: How is the "higher vertical mixing leading to similar concentrations" quantified? The yellow frame (15 to 17 July) shows also large differences in monoterpene concentrations at different heights.

- In this part, we are discussing measurements performed by both instruments at the same height, but at two different locations. This comparison includes a first period of measurements between the 13th and the 15th (green frame) and a second period

between the 17th and 18th of July (dashed green-yellow frame). During this second period, a higher vertical mixing is due to a higher u*, that was around 0.3 m s-1, higher to what was observed for stable nights (u*≈ 0.1 m s-1). The measurement period between the 15th and 17th of July corresponds to CRM and UL-FAGE measuring at two different heights.

New version, page 19, lines 547- 550 : Similar trends in OH reactivity are seen between the two datasets, even if the first period was associated with a clear vertical stratification (Fig. 4, green frame), leading to higher concentrations of monoterpenes within the canopy, whereas the second period was characterized by a higher vertical mixing (mean u* ≈ 0.3 m s-1), leading to similar concentrations of monoterpenes at the two heights (Fig. 4, dashed green-yellow frame).

8- P17 L21: (a) Which data are used for the linear regression discussed in this section?

- The data used for the regression with a slope of 1.22 and an intercept of -0.69 correspond to the period when LSCE-CRM and UL-FAGE measured at the same height but different horizontal locations (17th and 18th of July, dashed green-yellow frame in Fig. 4).

(b) It does not sound likely that inhomogeneities of air masses result in a change in the intercept, but would increase the scatter of data in the correlation.

New version: From the 13th to 15th midday of July (1st period) and from the 17th midday to 18th midday (2nd period), the two instruments were sampling at the same height but from different horizontal locations within the canopy (with sequential within/above canopy measurements for CRM during the second period). The horizontal distance between the two inlets was around 10 m as shown in Fig. 1. .... At the same height but different horizontal locations, the linear regression of LSCE- CRM data plotted against UL-FAGE data (not shown) indicates a good agreement with a slope of 1.22 ± 0.01, an intercept of -0.69 ± 0.17 and a correlation coefficient of 0.85 (1st and 2nd period). Compared to the results at the same location (vertical and horizontal), the slope and

the correlation coefficient are in the same range. Only the intercept differs significantly (-0.69 ± 0.17 compared to 4.22 ± 0.41). This change could be related to air mass inhomogeneities which could be systematically less reactive at one location compared to the other one. From these observations, we can conclude that reactivity measurements performed at different horizontal locations are consistent and that inhomogeneities in ambient air can lead to differences on the order of several s-1.

9- P18 L6: The reference Lou et al 2010 is not appropriate, because measurements in that paper were done in a mixed environment.

- The reference "Lou et al., 2010" was mentioned for the review part of it, in the introduction, where they put a table summarizing OH reactivity values in various environments. Instead, we now mention the review of "Yang et al., 2016" and "Dusanter and Stevens, 2017".

10- P18 L20 / P20 L22: The authors may want to mention already here that it is well known that plant emissions are increasing with increasing temperature.

New version P20, lines 596- 598: Another important parameter to consider is ambient temperature, which is known to enhance BVOCs emissions during the day when stomata are open, and which also plays a role for night-time emissions due to permeation, even though stomata are closed in the dark (Simon et al., 1994).

11- Section 3.3/3.5.: (a) The discussion would benefit, if the accuracy of calculated OH reactivity were taken into account (maybe also shown in Fig. 5).

The accuracy of calculated OH reactivity cannot be determined in a simple way and are rarely reported in previous studies. It depends on random (precision) and systematic (calibration) errors on trace gas measurements and errors on reported rate constants. Propagating the different types of errors (when known) is not straightforward. However, we can estimate it to be around 22-24%, as reported in Hansen et al. (2015). These values were obtained under similar experimental conditions than those used in

the Landex campaign, assuming that errors on rate constants are independent from each other and that errors on measured VOC concentrations are characterized by an independent random error of 5 % and a similar systematic error of 10 % for each VOC. This estimation has now been included in the revised version.

New version, page 23, lines 661- 662: Figure 6 shows that there is a good co-variation of the measured total OH reactivity by the CRM instrument with the values calculated from the PTR-MS data (22- 24% ($2\sigma$)).

(b) Is there an estimate of OH reactivity from oxidation products not taken into account here (for example from oxidation products like MVK/MACR)?

- As mentioned in page 14, lines 415- 420 (new version): "Since measurements from the PTR-MS instrument cover the whole campaign and were performed at the same heights than OH reactivity measurements, these measurements, including methanol, acetonitrile, acetaldehyde, acetone, isoprene, methacrolein + methylvinylketone + fragment ISOPOOH (MACR+MVK+ISOPOOH), methylethylketone (MEK) and the sum of monoterpenes (MTs), were selected to calculate the OH reactivity and to evaluate the potential missing OH reactivity at both levels". Oxidation products of isoprene were already taken into account in OH reactivity calculations. Regarding MTs oxidation products, their contribution to OH reactivity remains low (around 0.2 s-1 on average and a maximum of 1.2 s-1 together). However, and as reported in table 3 (new version), fragmentation was not corrected for and reported concentrations are likely lower limits.

The answer could be found in the new version of the paper:

Page 30, lines 826- 829: Checking monoterpenes' oxidation products variabilities (nopinone and pinonaldehyde), both nights exhibited higher concentration levels of these species, however their contribution to OH reactivity remained relatively low, and did not exceed 1 s-1, on average for both nights, keeping in mind that this is a lower limit of their contribution (since the reported measurements do not account for potential fragmentation in the PTR-MS).

[Figure]

(c) Is there any estimate, if transportation from other sources could have been impacted the location?

- We found some difference when checking air mass backward trajectories which suggested an explanation for the higher missing OH reactivity inside canopy for the 4th-5th, July night.

The answer could be found in the new version of the paper:

Page 30, line 832- Page 31, line 839, line: When looking at air masses backward trajectories (Fig. 10), the 4th-5th night was characterized by an air mass originally coming from the ocean, which spent at least 48 hours above the continent before reaching the site. This could have led to the enrichment of the air mass with species emitted by the widely spread Landes forests and their oxidation products. Thus, the significant missing OH reactivity observed during the mentioned night is likely related to unconsidered compounds of biogenic origin characterized by a similar behavior to that of isoprene, acetic acid and MVK+MACR+ISOPOOH, which accumulated in the stable nocturnal boundary layer. In contrast, air masses spent approximately 12-18 hours above the continent during the 6th-7th of July, with more time above the ocean. Marine air masses are generally known to be clean, with relatively low levels of reactive species.

12- Section 3.4.: Would the authors expect a difference in the distribution of OH reactants? Was there any attempt to estimate how much of the emissions were oxidized inside the canopy?

MVK+MACR+ISOPOOH/ isoprene had generally higher values during the day and were higher above the canopy, which suggests a difference in the distribution of OH reactants. Another paper on BVOCs reactivity with atmospheric oxidants (ozone, OH radical and nitrate) is in preparation. In this paper, differences of BVOCs consumption were observed between inside and above the canopy, which conducted to different distribution of co-reactants linked to difference of oxidants concentrations and/or BVOCs

concentrations between both levels ( Mermet et al., in preparation)

13- P29: Sesquiterpene oxidation products are likely not measured. Could the authors still estimate how much reactivity would be expected, if the difference between in and above canopy was due to oxidation?

It is mentioned in the new version of the paper, page 31, line 870- page 32, line 879:

Plotting the ratio SQT(above)/MTs(above) with the ratio SQT(inside)/MTs(inside) shows a good linear correlation with a slope of 0.72 and an R2 of 0.5. Knowing that sesquiterpenes are highly reactive with ozone (Ciccioli et al., 2002), which can dominate the chemistry during dark hours, this observation suggests that a larger fraction of these species (≈30%) could be consumed by ozonolysis above canopy, leading to the formation of unidentified secondary compounds. However, sesquiterpenes were present at relatively low concentrations (max of 0.25 ppbv and 0.12 ppbv, inside and above canopy, respectively). Assuming that all sesquiterpenes are b-caryophyllene and considering that 30% are transformed into first generation oxidation products through ozonolysis reactions, the maximum mixing ratio of these products would be around 0.07 ppb each assuming a yield of 1. However, it was reported by Winterhalter et al. (2009) that oxidation products of b-caryophyllene were much less reactive (100 times) than their precursor. Thus, the contribution of sesquiterpenes night-time oxidation products to the missing OH reactivity is likely negligible.

14- Figures in the main text and supplementary material: Font sizes are very small. It would be easier for the reader, if they were larger. The position of legend below the x-axis label is unusual. Errors bars of measurements would be helpful to judge differences, if quantities are compared.

All the suggestions of formatting have been taken into account.

Technical: The authors should follow the style of the journal for example how figures are referenced, dates are given and SI units should be used.

These points have been corrected.

References: - ACTRIS. 2014. "WP4- NA4: Trace Gases Networking: Volatile Organic Carbon and Nitrogen Oxides Deliverable D4.9: Final SOPs for VOCs Measurements." ACTRIS.

- S. Kim, T. Karl, D. Helmig, R. Daly, R. Rasmussen, and A. Guenther, Atmos. Meas. Tech., 2, 99–112, https://doi.org/10.5194/amt-2-99-2009, 2009.

- Yang, Y., Shao, M., Wang, X., Nölscher, A. C., Kessel, S., Guenther, A. and Williams, J.: Towards a quantitative understanding of total OH reactivity: A review, Atmos. Environ., 134(2), 147–161, doi:10.1016/j.atmosenv.2016.03.010, 2016.

- S. Dusanter and P. Stevens, Recent Advances in the Chemistry of OH and HO2 Radicals in the Atmosphere: Field and Laboratory Measurements, Advances in Atmospheric Chemistry, pp. 493-579 (2017).

- K. Mermet, E. Perraudin, S. Dusanter, S. Sauvage, T. Léornadis, P.-M. Flaud, S. Bsaibes, J. Kammer, V. Michoud, A. Gratien, M. Cirtog, M. Al Ajami, F. Truong, S. Batut, C. Hecquet, J.-F. Doussin, C. Schoemaecker, V. Gros, N. Locoge and E. Villenave, Atmospheric Reactivity of Biogenic Volatile Organic Compounds in a Maritime Pine Forest during the LANDEX Field Campaign, in preparation.

- Winterhalter, R., Herrmann, F., Kanawati, B., Nguyen, T. L., Peeters, J., Vereecken, L. and Moortgat, G. K.: The gas-phase ozonolysis of $\beta$-caryophyllene (C15H 24). Part I: An experimental study, Phys. Chem. Chem. Phys., 11(21), 4152–4172, doi:10.1039/b817824k, 2009.

Please also note the supplement to this comment:
https://www.atmos-chem-phys-discuss.net/acp-2019-548/acp-2019-548-AC2-supplement.pdf
* * *
[Figure]

2019.

**Supplement:**

**Table 2.** Summary of corrections applied to raw reactivity data for LSCE-CRM. Correction coefficients are obtained from experiments performed before, during and after the field campaign.

| Correction | Correction factor | Mean absolute change in OH reactivity ($s^{-1}$) |
|---|---|---|
| **Humidity changes between C2 & C3** | -89.18±2.16 | + 2.4 |
| **Not operating the CRM under pseudo first order conditions** | F= (-0.52±0.20)×(pyrrole-to-OH)+ (3.38±0.60) | + 10.0 |
| **Dilution** | D= 1.46 | + 2.8 |

[Figure]

**Figure 1.** Measured reactivity by LSCE- CRM instrument as function of the measured reactivity by UL- FAGE when both instruments were measuring at the same location within the canopy (data resampled with a time resolution of 1 min). Errors bars represent the overall systematic uncertainty ($1\sigma$) that is around 15 % and 35 % for LP- LIF and the CRM, respectively.

---

## Author Response (AR1)

**Publication: acp-2019-548**

**Title: Variability of OH reactivity in the Landes maritime Pine forest: Results from the LANDEX campaign 2017**

Dear Co-Editor,

Please find enclosed the revised version of the manuscript entitled "Variability of OH reactivity in the Landes maritime Pine forest: Results from the LANDEX campaign 2017". The work presented, part of the LANDEX- 2017 campaign, aimed to characterize the variability of BVOCs and their OH reactivity inside and above the canopy of a maritime Pine forest, South of France, during July 2017. This version includes most of the suggestions made by the reviewers, as explained in the author's response, which detailed point by point our answers to all reviewers comments.

Please note that, between the submission of the replies to the reviewers comments and the preparation of this new version, minor corrections were made, which only very slightly affected some of the values (most of the time, it only concerns the first digit after the decimal point) mentioned in the responses to referees comments or in the initial manuscript. It is important to note that, these corrections did not induce any change in the results interpretation and in the conclusion of this study. For clarity, a list of these minor changes is reported after the responses to the reviewers and before the revised manuscript.

As the manuscript has been significantly improved, thanks to all reviewers comments, we hope that it has now all the elements to be accepted in ACP.

We thank you for your consideration,

Kind Regards,

Sandy BSAIBES

- **Answers to RC1**

1- Abstract lines 29-30: Could you also add a comment or a value how big fraction was missing?

Revised version, page 1, lines 31- 32: Comparing the measured and the calculated OH reactivity highlighted an average missing OH reactivity of 22% and 33%, inside and above the canopy, respectively.

2- Page 6, lines 15-20: How about O3? Did you apply any $O_3$ correction? Have you detected any effect of O3 in your CRM system?

- Based on previous experiments (Fuchs et al., 2017), no ozone dependency was seen for the LSCE-CRM. Therefore, no tests were performed to characterize the interference due to $O_3$ and no correction was applied to OH reactivity raw data.

This information has been added in the revised version of the manuscript as:

Revised version, page 6, lines 176- 179: In some CRM systems, corrections for potential $NO_2$ and/or $O_3$ artefacts are also considered (Michoud et al., 2015, Praplan et al., 2017). On one hand, NO2 is subject to photolysis leading to NO, which can subsequently react with $HO_2$ yielding OH. On the other hand, $O_3$ can also be photolyzed in the reactor, producing $O(^1D)$, which reacts further with $H_2O$, yielding two OH radicals.

And page 8, lines 228 -232: NO mixing ratios were lower than 0.5 ppbv (corresponding to the detection limit of the NOx monitor deployed during LANDEX) most of the time for the measurement time periods used in this study, and no correction was applied for the spurious formation of OH from the $HO_2$+NO reaction. Similarly, for $NO_2$, no correction was applied due to the low ambient mixing ratio of 1.1 ± 0.8 ppbv. Regarding $O_3$, no dependency was seen for LSCE-CRM, based on previous experiments (Fuchs et al., 2017). Therefore, no correction was applied. The correction (D) on the reactivity values due to the dilution was around 1.46 during the campaign.

3- Page 7, lines 27-28: Please, be more specific. What was the concentration range of isoprene and a-pinene?

Revised version, page 7, lines 221 - 223: To determine the correction factor for the deviation from pseudo-first order kinetics, injections of known concentrations of isoprene ($k_{isoprene+OH}$ = $1x10^{-10}$ $cm^3$ molecule$^{-1}$ s$^{-1}$, 1- 120 ppbv) and α-pinene ($k_{α-pinene+OH}$ = 5.33 x $10^{-11}$ $cm^3$ molecule$^{-1}$ s$^{-1}$, 3 -190 ppbv) (Atkinson, 1985) were performed before and after the field campaign since they represent the dominant species in this forest ecosystem.

4- Page 9, lines 28-29: (a) Copper tubing impregnated with KI is commonly used for the DNPH measurements of aldehydes and ketones, but is it suitable for monoterpenes? Did you test the recovery of terpenes?

- As presented in Mermet et al. 2019 (AMTD), several tests were performed on scrubbers recommended by ACTRIS (copper tubes coated with potassium iodide, glass filters impregnated with sodium thiosulfate, and copper screens coated with manganese dioxide) to characterize (1) $O_3$ removal efficiency, (2) losses of BVOCs in the absence of ozone, and (3) potential ozone-induced losses of BVOCs in the scrubber. Copper tubes coated with potassium iodide (KI) appeared as the best choice for BVOC measurements. In the absence of ozone, KI scrubbers exhibited BVOC losses lower than 5 % for most non-oxygenated species, whereas in the presence of ozone, losses were relatively higher but remained lower than 15% (lower than 5 % for α- and β-pinene). The only two notable exceptions

were the most reactive compounds, i.e. α-terpinene and β-caryophyllene, whose losses were approximately 20 % and 40 %, respectively. These two species represent only a minor fraction (3 % maximum) of the total sum of compounds measured with GC-BVOC2 inside the canopy, compared to maxima of 42- 43 % for α and β-pinene.

(b) What about particle filter? Do you see losses of terpenes in them?

- No tests were made on the particle filters. ACTRIS 2014 measurement guidelines were followed. High flow rates were set in the sampling lines: 1 L min$^{-1}$ for GC instruments and 10 L min$^{-1}$ for the PTR-MS. The contact time between ambient BVOCs and the particle filters is extremely short and we don't expect significant losses.

(c) Maybe you could provide some reference on an earlier study where they have been tested.

- ACTRIS. 2014. "WP4- NA4: Trace Gases Networking: Volatile Organic Carbon and Nitrogen Oxides Deliverable D4.9: Final SOPs for VOCs Measurements." ACTRIS.

This information has been added in the revised version of the manuscript as:

Revised version Page 10, lines 304- 311: Measurements of VOCs (Table 3) were performed at different locations (Fig. 1) by a proton transfer reaction-mass spectrometer (PTR-MS) and four on-line gas chromatographic (GC) instruments. Ozone scrubbers (Copper tube impregnated with KI) and particle filters were added to the inlets of all GC sampling lines. Losses of BVOCs in these ozone scrubbers were investigated under similar sampling conditions in the absence and presence of O$_3$ (Mermet et al., 2019, AMTD). The scrubbers exhibited less than 5 % losses for most non-oxygenated BVOCs, whereas in the presence of ozone, losses were relatively higher for some BVOCs, but remained lower than 15 % (lower than 5 % for α- and β-pinene). High flow rates were applied in the sampling lines: 1 L min$^{-1}$ for GC instruments and 10 L min$^{-1}$ for the PTR-MS, therefore, the contact time between ambient BVOCs and the particle filters was extremely short and no significant losses are expected.

5- Page 10, line 2 and 14: You used Carbotrap B and C for collecting terpenes. I am worried that they are not very good for mono- and sesquiterpenes and you may have some losses of them? Did you do some recovery tests? Have you detected any losses or isomerization while testing those? I would recommend for example Tenax TA cold trap for mono- and sesquiterpenes.

- Carbotrap C in GC-BVOC1 is already set by the manufacturer. Carbotrap B has been selected among the possible adsorbent as listed in the ACTRIS guidelines (ACTRIS, 2014). The method has been optimized in terms of temperature of the thermodesorption, the column, the sampling volume and sampling line including a scrubber. Results are shown in the Mermet et al. AMTD, 2019. Based on a reference mixture composed of 14 monoterpenes, tests resulted in a good separation for most of the compounds. Apart sabinene and terpinene, a good recovery has been obtained between the experimental response coefficient compared to the theoretical ones (determined from the Equivalent carbon number for FID). As a consequence, the calculated uncertainties are significantly higher for these 2 compounds, for which some isomerization or thermodegradation could occur. Indeed, Tenax TA is another well characterized adsorbent but thermodegradtion of monoteprenes may also occur as reported by Coeur et al. (1997).

This information has been added in the revised version of the manuscript as:

Revised version page 11, lines 333- 335: The method has been optimized in terms of temperature of the thermodesorption, the column, the sampling volume and sampling line including a scrubber. More details about the optimization and the tests performed can be found in Mermet et al. AMTD, 2019.

6- Page 10, line 12: In some of the MARKES Unity systems b-pinene and some other monoterpenes are isomerized and concentrations of some monoterpenes, for example p-cymene, are increasing over the time. Did you detect low response for b-pinene or for some other monoterpenes or increase of p-cymene?

- p-cymene response observed was elevated comparing to other monoterpenes. For some monoterpenes a low response was observed. It is the case of sabinene, terpinolene, 2-carene for example, but not for the most abundant monoterpenes such as b-pinene, a-pinene, limonene or myrcene (Mermet et al., 2019). While isomerization may be an issue for measuring some monoterpenes with this instrument, the most abundant contributors to the OH reactivity are well measured and this issue does not impact the conclusions of this study. The method could be optimized by using another desorption system.

To take into account, the question of the reviewer, in the revised manuscript we refer the reader to the paper of Mermet et al. which gives all the results concerning the optimization and the tests which have been performed.

This has been added to the revised version of the paper page 11, lines 334- 336: Tests showed a low response for some compounds (i.e. sabinene, terpinolene, …), however, the most abundant compounds, were well measured. More details about the optimization and the tests performed can be found in Mermet et al. (2019, AMTD).

7- Section 3.3.: Was the mean missing fraction higher inside or above canopy? I would guess there are more reaction products above the canopy.

- Section 3.3 aims to present a comparison between measured and calculated OH reactivity whereas missing reactivity (as absolute and relative fractions) is discussed in section 3.5. The mean relative missing fraction was around 48 % above the canopy and 38 % inside the canopy, when comparing the measured OH reactivity with the calculated one from PTR-MS data, which was measuring at both heights. However, it should be reminded that, measurements were not performed simultaneously above and inside the canopy, except for a short period from mid-day of the 17th, July to mid-day of the 18th, July.

This information is mentioned in the text:

Revised version page 28, lines 748- 749: When comparing measurements of OH reactivity with calculations based on PTR-MS data (see Table 3), an average of 38% (7.3 s$^{-1}$) and 48% (6.0 s$^{-1}$), remained unexplained inside and above the canopy, respectively.

8- Page 26, line 12: Is the typical B-value (0.057) for the monoterpene emissions or for the reactivity? Often B-value 0.09 is used for the monoterpene emissions.

- The β value is normally used for monoterpenes emissions from vegetation. When applied on missing OH reactivity data, it can be used to indicate if the missing OH reactivity is linked to primary emissions that are temperature-dependent like monoterpenes. When the measured ROH was compared to the calculated one from PTR-MS data, a β of 0.09 was obtained when the missing ROH was fitted in the equation used to describe the temperature dependency of monoterpenes emissions. This β was in the range of β-values normally seen for monoterpenes emissions. However, following the remark of reviewer 3, we have decided to examine the missing reactivity by taking into account in the calculated reactivity all the measured compounds available at the 6 m height. In this case, the missing was also fitted in the exponential relation, but the β value was higher (0.17), which indicates that the missing fraction is not only linked to primary

emissions but is also due to secondary oxidation products (Mao et al., 2012, Hansen et al. 2014, Kaiser at al.,2016).

Revised version, page 29, lines 775- 789: As reported in Di Carlo et al. (2004), the missing OH reactivity was fitted with an equation usually used to describe temperature dependent emissions of monoterpenes (Guenther et al., 1993): E(T) = E (293) exp(β(T293)), where E(T) and E(293) represent the emission rate at a given temperature T and at 293 K, respectively. In this equation, E(T) was substituted to MROH(T) and E (293) by MROH (293) with MROH representing the missing OH reactivity (Hansen et al., 2014). The value of β determined from the fit of the data for the 6 m height (day-time), is around 0.17, higher than the values attributed to monoterpenes emissions from vegetation (0.057 to 0.144 $K^{-1}$). Higher β-values were also obtained by Mao et al. (2012), Hansen et al. (2014) and Kaiser et al. (2016), were they suggested that daytime missing reactivity is mostly linked to secondary oxidation products. However, the use of β factor must be made with caution, as the missing OH reactivity can be influenced by processes that do not affect BVOCs emissions (i.e. the boundary layer height and the vertical mixing). Furthermore, we cannot exclude the possibility of light and temperature dependent emissions. Indeed, Kaiser et al. (2016) also investigated the temperature dependency of day-time missing OH reactivity in an isoprene-dominated forest, reporting that part of the missing emissions could be characterized by a light and temperature dependence, knowing that temperature increases with increasing solar radiation. Regarding above canopy, most measurements were performed during cool days. Thus, it was not possible to analyze the temperature dependence of above canopy day-time missing OH reactivity.

9- Page 31, 14-15: I think that also for monoterpenes reactions with ozone can be very significant. Do you have any idea of OH radical concentrations at the site? It would be nice to know how much lower the lifetimes of VOCs were during the day and how important ozone reactions were. Sometimes ozone reactions can be very important also during the day.

Based on the referee's comment, calculations of α-pinene lifetime (one of the major compounds) towards OH and O3 were made.

Information has been added in the new version of the paper, page 26, lines 723- 734:

The concentration of OH was $4.2 \times 10^6$ molecules $cm^{-3}$ on average during day-time with a maximum of $4.3 \times 10^7$ molecules $cm^{-3}$ and around $1.5 \times 10^6$ molecules $cm^{-3}$ on average during night-time (data available between the 13th and the 19th, July). However, a potential artefact on OH radical's measurements leading to a possible overestimation of OH radical's concentrations, could not be ruled out. Regarding ozone, its mixing ratio showed a diurnal cycle with maximum values during the day (max ≈ 60 ppbv, mean ≈ 29 ppbv), that were similar within and above the canopy due to efficient mixing, and lower levels during nights, with an average of 18 ppbv inside canopy, while levels higher by 1 - 9 ppbv on average, above the canopy. Considering OH and $O_3$ average mixing ratios, the α-pinene lifetime was estimated to be 1.2 hours and 4 hours, respectively, during the day, and 3.6 hours and 5.8 hours, respectively, during the night. At maximum OH and $O_3$ mixing ratios during day-time, the α-pinene lifetime was reduced to 7.4 min and 2 hours, respectively. Thus, OH chemistry remained dominant compared to ozonolysis of main emitted compounds on this site (i.e. α-pinene). An article on the reactivity of monoterpenes with OH, ozone and nitrate for this campaign is in preparation (Mermet et al., in preparation).

**Technical comments:**

10- Table 1: Please, add an explanation to K' max
-    Revised version (Table 1): ROH max ($s^{-1}$) instead of K' max ($s^{-1}$).

11- Page 10, line 13: You mention B-caryophyllene here, but it is not included into table 2. It should be removed from the text.
-   B-caryophyllene was added in Table 3 of the revised paper.

12- You have lots of time series plots, but they are a bit hard to follow and it would be also nice to get some quick and easy to look at average plots or tables (for example mean reactivity and mean missing reactivity during night and day, inside and above canopy and during cold and warm nights).

-   A table has been added in the revised version of the paper, page 28:

Table 4. Summary of the measured OH reactivity and the missing OH reactivity inside and above the canopy, during the day and the night, taking into account only PTR-MS data or all the data available at each height for OH reactivity calculations. These averages are calculated for the periods when CRM, PTR-MS and others instruments data are available.

|  | Mean Measured OH reactivity ($s^{-1}$) | Mean missing OH reactivity with PTRQi-ToFMS ($s^{-1}$) | Missing ROH considering PTRQi-ToFMS data + other measurements ($s^{-1}$) |
|---|---|---|---|
| **Inside** | 19.0 | 7.3 | 4.3 |
| Day | 18.4 | 7.0 | 4.1 |
| Night | 21.4 | 9.0 | 5.6 |
| Stable cool nights | 20.5 | 5.7 | 2.1 |
| Stable warm nights | 41.6 | 10.9 | 6.9 |
| Unstable cool nights | 7.9 | 4.5 | <LOD |
| Unstable warm nights | 13.5 | 6.8 | 3.6 |
| **Above** | 12.6 | 6.0 | 4.2 |
| Day | 10.4 | 5.0 | 3.1 |
| Night | 15.5 | 7.5 | 5.6 |
| Stable cool nights | 14.8 | 7.5 | 5.7 |
| Stable warm nights | ____ | ____ | ____ |
| Unstable warm nights | 20.5 | 7.1 | 5.2 |
| Unstable cool nights | ____ | ____ | ____ |

- A more detailed table has been added in the supplementary material: Table S9

13- Page 28, line 16: ')' is missing.

- Corrected.

14- Page 28, line 21: Should this be 'This compound showed a diurnal cycle similar to that of isoprene (Fig 4.c) and was not used to calculate . . .'?

-   Indeed.

Revised version, page 30, lines 810- 811: This compound showed a diurnal cycle similar to that of isoprene (Fig 5.c) and was not used to calculate the OH reactivity.

15- Page 29, line 5: What is '(S9)'?

- S9 is supplementary material 9.

16- Page 31, lines 8-10: I did not understand this sentence 'Complementary measurements performed inside (O3, NOx) and above the canopy (OVOCs, NMHCs, O3, NOx and butanol), explained with methane and carbon monoxide, part of the missing OH reactivity, that remained significant for warm days and stable/ warm nights.'

This part of the conclusion was modified: An investigation of the missing OH reactivity indicated averages of 6.0 and 7.3 $s^{-1}$ inside and above the canopy, respectively, over the whole campaign. However, it showed some diurnal variability at both heights. During day-time, higher missing OH reactivity was observed on warmer days inside and above the canopy. Plotted against temperature, inside canopy missing OH reactivity showed a dependency on temperature. The analysis suggested that the missing OH reactivity may be due to unmeasured primary emitted compounds and oxidation products. In this context, OH reactivity measurements from a Pinus pinaster Aiton branch enclosure, could be of great interest to verify the contribution of unaccounted/unmeasured BVOCs emissions to OH reactivity as done by Kim et al. (2011), for red oak and white pine branch enclosures. Furthermore, higher levels of isoprene oxidation products on warmer days also suggest that the missing reactivity could be due to the formation of unmeasured oxidation products. Regarding the night-time period, the highest missing OH reactivity was found inside canopy for the 4th-5th, July night. This night was characterized by higher levels of isoprene and its oxidation products, compared to the night of the 6th-7th, July with similar atmospheric conditions. Air masses backward trajectories showed a continental origin for this night, suggesting that species, emitted by the largely spread Landes forest, could have been imported to the site and accumulated due to the stable nocturnal boundary layer. These species, unmeasured by the deployed analytical instruments and hence not considered in OH reactivity calculations, could explain the higher missing OH fraction for the 4th-5th, July night. Finally, the investigation of sesquiterpenes and monoterpenes oxidation products (nopinone and pinonaldehyde) measured by PTR-MS highlighted their small contribution in terms of OH reactivity. They only explained a small fraction of the observed missing OH reactivity inside and above canopy during night.

**References:**

- Mermet, K., Sauvage, S., Dusanter, S., Salameh, T., Léonardis, T., Flaud, P.-M., Perraudin, É., Villenave, É., and Locoge, N.: Optimization of a gas chromatographic unit for measuring BVOCs in ambient air, Atmos. Meas. Tech. Discuss., https://doi.org/10.5194/amt-2019-224, in review, 2019
- ACTRIS, 2014. WP4- NA4: Trace gases networking: Volatile organic carbon and nitrogen oxides Deliverable D4.9: Final SOPs for VOCs measurements. ACTRIS.
- Coeur, C., Jacob, V., Denis, I., Foster, P., 1997. Decomposition of α-pinene and sabinene on solid sorbents, tenax TA and carboxen. J. Chromatogr. A 786, 185–187. https://doi.org/10.1016/S0021-9673(97)00562-1
- Atmospheric Reactivity of Biogenic Volatile Organic Compounds in a Maritime Pine Forest during the LANDEX Field Campaign Kenneth Mermet, Emilie Perraudin, Sébastien Dusanter, Stéphane Sauvage, Thierry Léornadis, Pierre-Marie Flaud, Sandy Bsaibes, Julien Kammer, Vincent Michoud, Aline Gratien, Manuela Cirtog, Mohamad Al Ajami, François Truong, Sébastien Batut, Christophe Hecquet, Jean-Francois Doussin, Coralie Schoemaecker, Valérie Gros, Nadine Locoge and Eric Villenave, in preparation.

- **Answers to RC2:**

1- The characterization experiments for the CRM are described, but it remains unclear, how large corrections were. The authors should consider give some numbers, how big corrections were for typical chemical conditions of this campaign. A discussion about consequences for the accuracy of measurements would be beneficial.

Revised version, page 8, lines 237 - 240: Table 2 reports a summary of the corrections resulting from our tests and their impact on measurements. As shown in table 2, the application of (F), for the deviation from pseudo-first order kinetics, induces the largest correction, with an absolute increase of 10.02 $s^{-1}$ on average. Furthermore, this factor (F) has the largest relative uncertainty, with ±36 %, against ±2 % for the humidity correction factor.

| Correction | Correction factor | Mean absolute change in OH reactivity ($s^{-1}$) |
|---|---|---|
| Humidity changes between C2 and C3 | -89.18±2.16 | + 2.4 |
| Not operating the CRM under pseudo first order conditions | F = (-0.52±0.20)×(pyrrole-to-OH)+ (3.38±0.60) | + 10.0 |
| Dilution | D = 1.46 | + 2.8 |

2- The authors mention that one of the conclusions from previous campaigns were that potential loss of reactive VOCs could be a problem in CRM instruments. Did they quantitatively test this for example when they did the characterization experiment for the deviation from a pseudo-first order reaction system?

- In order to minimize potential losses of reactive VOCs in the CRM sampling system, heated (≈50 °C) sulfinert lines were used. Indeed, Kim et al. (2009), showed that losses of β-caryophyllene are negligible in heated lines with temperatures above 20 °C. More details are also mentioned in the answer to comment 3.

Information has been added in the revised version of the manuscript as:

Page 7, line 199- 201: Ambient air was sampled through two 1/8" OD sulfinert lines, collocated on a mast close to the trailer (see Fig. 1(a)). The lines lengths were 8 m for the measurements performed inside the canopy and 12 m for those performed above. These lines were heated up to 50 °C as it was shown that losses of highly reactive molecules (β-caryophyllene) were negligible for temperature above 20 °C (Kim et al., 2009).

3- Similarly, did the authors test, if VOCs were quantitatively transmitted through inlet lines for the GC and PTR-MS analysis? How often were filters in inlet lines exchanged and did they authors test, if the transmission of VOC through filters decreased with time?

- For GC instruments, VOCs were sampled through sulfinert sampling lines, similar to those used in the CRM sampling system, heated up to 50 °C, with a flow rate of, at least, 1 L $min^{-1}$, ensuring a short residence time of less than 8 s. Materials used are recommended by ACTRIS guidelines (ACTRIS 2014). Regarding the PTR-MS, the sampling lines were made of PFA (1/4"-OD) and were heated at 50 °C. All lines were 15-m long and the flow rates were adjusted to 10 L $min^{-1}$ to reduce the residence time below 2-s. Filters were also made of PFA and were changed every 2-weeks. No tests were performed to check the transmission of VOCs. However, Kim et al. (2009)

tested losses of β-caryophyllene (a sticky sesquiterpene) in a 40-m long Teflon tube (1/4"-OD) flushed at 25 L min⁻¹. These operating conditions lead to a residence time similar to that observed during LANDEX for our PTR-MS sampling system. The authors varied the line temperature from zero to 40 °C using a temperature controlled environmental chamber and showed that losses of β-caryophyllene are negligible above 20 °C. The PTR-MS lines being heated to 50 °C in this study, no losses are expected for VOCs reported in this study.

This information has been added in the revised version of the manuscript as:

Page 12, lines 356- 359: Sulfinert material chosen for all GCs sampling lines and used in LSCE-CRM sampling system, is recommended by ACTRIS 2014. High flows were set in the lines (residence time of less than 8 s), that were heated up to 50 °C to minimize the losses of potential reactive species. Filters and scrubbers were changed twice for the GC-BVOC1 and one time for the other GC instruments.

Page 12, lines 364- 366: The lines (PFA, 1/4" OD) were heated at 50 ∘C and constantly flushed at 10 L min⁻¹ using an additional pump and rotameters. Indeed, Kim et al. (2009) tested losses of β-caryophyllene in similar operating conditions. Authors varied the temperature from zero to 40 °C showing that losses of b-caryophyllene are negligible above 20 °C. The residence time was lower than 2s.

4- The authors should mention early in the paper, how they deal with contributions of NO2 / NO to the OH reactivity.

Revised version page 16, lines 469- 474: A large range of NMHCs and OVOCs were measured at the 12 m height only by GC-NMHC and GC-OVOC (Table 3). Butanol (from SMPS exhausts) was also checked and found to be negligible at 12 m and highly and rapidly variable at 6 m (short peaks). NO and $NO_2$ were only measured at the 6 m height. Mean NO mixing ratio was below the LOD for the measurement period and $NO_2$ was around 1.1 ± 0.8 ppbv on average. Thus, it was chosen not to take these species into account in the OH reactivity calculations, since they are not available at both levels. However, sensitivity tests were performed, in order to compute the relative contribution of butanol, OVOCs and NMHCs to OH reactivity (See section 3.5 and Fig. S5 and S6).

5- Page 14 Point 3). It would be useful to give some numbers for the estimate of OH reactivity from species only measured at 12m height in the main text.

- This paragraph (point 3, page 16 of the revised paper) describes the methodology used. No results were included. The contribution of species only measured at 12m to OH reactivity is mentioned on page 28 of the revised version, when investigating the missing OH reactivity.

6- Figure 3: In a correlation plot, error bars of measurements are needed. Did the regression procedure take into account errors of the measurements?

Errors bars were added as shown in Fig. 3, page 18 of the revised version. Errors of the measurements were not taken into account in the regression procedure.

[Figure]

**Figure 1.** Measured reactivity by LSCE- CRM instrument as function of the measured reactivity by UL- FAGE when both instruments were measuring at the same location within the canopy (data resampled with a time resolution of 1 min). Errors bars represent the overall systematic uncertainty (1σ) that is around 15 % and 35 % for LP- LIF and the CRM, respectively.

7- P17 L19: How is the "higher vertical mixing leading to similar concentrations" quantified? The yellow frame (15 to 17 July) shows also large differences in monoterpene concentrations at different heights.

- In this part, we are discussing measurements performed by both instruments at the same height, but at two different locations. This comparison includes a first period of measurements between the 13th and the 15th (green frame) and a second period between the 17th and 18th of July (dashed green-yellow frame). During this second period, a higher vertical mixing is due to a higher u*, that was around 0.3 m s$^{-1}$, higher to what was observed for stable nights (u*≈ 0.1 m s$^{-1}$). The measurement period between the 15th and 17th of July corresponds to CRM and UL-FAGE measuring at two different heights.

Revised version, page 19, lines 547- 550: Similar trends in OH reactivity are seen between the two datasets, even if the first period was associated with a clear vertical stratification (Fig. 4, green frame), leading to higher concentrations of monoterpenes within the canopy, whereas the second period was characterized by a higher vertical mixing (mean u* ≈ 0.3 m s$^{-1}$), leading to similar concentrations of monoterpenes at the two heights (Fig. 4, dashed green-yellow frame).

8- P17 L21: (a) Which data are used for the linear regression discussed in this section?

The data used for the regression with a slope of 1.22 and an intercept of -0.69 correspond to the period when LSCE-CRM and UL-FAGE measured at the same height but different horizontal locations (17th and 18th of July, dashed green-yellow frame in Fig. 4).

(b) It does not sound likely that inhomogeneities of air masses result in a change in the intercept, but would increase the scatter of data in the correlation.

Revised version, page 18: From the 13th to 15th midday of July (1st period) and from the 17th midday to 18th midday (2nd period), the two instruments were sampling at the same height but from different horizontal locations within the canopy (with sequential within/above canopy measurements for CRM during the second period). The horizontal distance between the two inlets was around 10 m as shown in Fig. 1. .... At the same height but different horizontal locations, the linear regression of LSCE- CRM data plotted against UL-FAGE data (not shown) indicates a good agreement with a slope of $1.22 \pm 0.01$, an intercept of $-0.69 \pm 0.17$ and a correlation coefficient of 0.85 (1st and 2nd period). Compared to the results at the same location (vertical and horizontal), the slope and the correlation coefficient are in the same range. Only the intercept differs significantly ($-0.69 \pm 0.17$ compared to $4.22 \pm 0.41$). This change could be related to air mass inhomogeneities which could be systematically less reactive at one location compared to the other one. From these observations, we can conclude that reactivity measurements performed at different horizontal locations are consistent and that inhomogeneities in ambient air can lead to differences on the order of several $s^{-1}$.

9- P18 L6: The reference Lou et al 2010 is not appropriate, because measurements in that paper were done in a mixed environment.

- The reference "Lou et al., 2010" was mentioned for the review part of it, in the introduction, where they put a table summarizing OH reactivity values in various environments. Instead, we now mention the review of "Yang et al., 2016" and "Dusanter and Stevens, 2017".

10- P18 L20 / P20 L22: The authors may want to mention already here that it is well known that plant emissions are increasing with increasing temperature.

Revised version page 20, lines 596- 598: Another important parameter to consider is ambient temperature, which is known to enhance BVOCs emissions during the day when stomata are open, and which also plays a role for night-time emissions due to permeation, even though stomata are closed in the dark (Simon et al., 1994).

11- Section 3.3/3.5.: (a) The discussion would benefit, if the accuracy of calculated OH reactivity were taken into account (maybe also shown in Fig. 5).

- The accuracy of calculated OH reactivity cannot be determined in a simple way and are rarely reported in previous studies. It depends on random (precision) and systematic (calibration) errors on trace gas measurements and errors on reported rate constants. Propagating the different types of errors (when known) is not straightforward. However, we can estimate it to be around 22-24%, as reported in Hansen et al. (2015). These values were obtained under similar experimental conditions than those used in the Landex campaign, assuming that errors on rate constants are independent from each other and that errors on measured VOC concentrations are characterized by an independent random error of 5 % and a similar systematic error of 10 % for each VOC. This estimation has now been included in the revised version.

Revised version, page 23, lines 661- 662: Figure 6 shows that there is a good co-variation of the measured total OH reactivity by the CRM instrument with the values calculated from the PTR-MS data (22- 24% (2σ)).

(b) Is there an estimate of OH reactivity from oxidation products not taken into account here (for example from oxidation products like MVK/MACR)?

As mentioned in page 14, lines 415- 420 (revised version): "Since measurements from the PTR-MS instrument cover the whole campaign and were performed at the same heights than OH reactivity measurements, these measurements, including methanol, acetonitrile, acetaldehyde, acetone, isoprene, methacrolein + methylvinylketone + fragment ISOPOOH (MACR+MVK+ISOPOOH), methylethylketone (MEK) and the sum of monoterpenes (MTs), were selected to calculate the OH reactivity and to evaluate the potential missing OH reactivity at both levels". Oxidation products of isoprene were already taken into account in OH reactivity calculations. Regarding MTs oxidation products, their contribution to OH reactivity remains low (around $0.2$ s$^{-1}$ on average and a maximum of $1.2$ s$^{-1}$ together). However, and as reported in table 3 (new version), fragmentation was not corrected for and reported concentrations are likely lower limits.

The answer could be found in the new version of the paper:

Page 30, lines 826- 829: Checking monoterpenes' oxidation products variabilities (nopinone and pinonaldehyde), both nights exhibited higher concentration levels of these species, however their contribution to OH reactivity remained relatively low, and did not exceed $1$ s$^{-1}$, on average for both nights, keeping in mind that this is a lower limit of their contribution (since the reported measurements do not account for potential fragmentation in the PTR-MS).

(c) Is there any estimate, if transportation from other sources could have been impacted the location?

We found some difference when checking air mass backward trajectories which suggested an explanation for the higher missing OH reactivity inside canopy for the 4th-5th, July night.

The answer could be found in the revised version of the paper:

Page 30, line 832- Page 31, line 839, line: When looking at air masses backward trajectories (Fig. 10), the 4th-5th night was characterized by an air mass originally coming from the ocean, which spent at least 48 hours above the continent before reaching the site. This could have led to the enrichment of the air mass with species emitted by the widely spread Landes forests and their oxidation products. Thus, the significant missing OH reactivity observed during the mentioned night is likely related to unconsidered compounds of biogenic origin characterized by a similar behavior to that of isoprene, acetic acid and MVK+MACR+ISOPOOH, which accumulated in the stable nocturnal boundary layer. In contrast, air masses spent approximately 12-18 hours above the continent during the 6th-7th of July, with more time above the ocean. Marine air masses are generally known to be clean, with relatively low levels of reactive species.

12- Section 3.4.: Would the authors expect a difference in the distribution of OH reactants? Was there any attempt to estimate how much of the emissions were oxidized inside the canopy?

- MVK+MACR+ISOPOOH/ isoprene had generally higher values during the day and were higher above the canopy, which suggests a difference in the distribution of OH reactants. Another paper on BVOCs reactivity with atmospheric oxidants (ozone, OH radical and nitrate) is in preparation. In this paper, differences of BVOCs consumption were observed between inside and above the canopy, which conducted to different distribution of co-reactants linked to difference of oxidants concentrations and/or BVOCs concentrations between both levels (Mermet et al., in preparation).

13- P29: Sesquiterpene oxidation products are likely not measured. Could the authors still estimate how much reactivity would be expected, if the difference between in and above canopy was due to oxidation?

It is mentioned in the revised version of the paper, page 31, line 870- page 32, line 879: Plotting the ratio SQT(above)/MTs(above) with the ratio SQT(inside)/MTs(inside) shows a good linear correlation with a slope of 0.72 and an R2 of 0.5. Knowing that sesquiterpenes are highly reactive with ozone (Ciccioli et al., 2002), which can dominate the chemistry during dark hours, this observation suggests that a larger fraction of these species (≈30%) could be consumed by ozonolysis above canopy, leading to the formation of unidentified secondary compounds. However, sesquiterpenes were present at relatively low concentrations (max of 0.25 ppbv and 0.12 ppbv, inside and above canopy, respectively). Assuming that all sesquiterpenes are b-caryophyllene and considering that 30 % are transformed into first generation oxidation products through ozonolysis reactions, the maximum mixing ratio of these products would be around 0.07 ppbv each assuming a yield of 1. However, it was reported by Winterhalter et al. (2009) that oxidation products of β-caryophyllene were much less reactive (100 times) than their precursor. Thus, the contribution of sesquiterpenes night-time oxidation products to the missing OH reactivity is likely negligible.

14- Figures in the main text and supplementary material: Font sizes are very small. It would be easier for the reader, if they were larger. The position of legend below the x-axis label is unusual. Errors bars of measurements would be helpful to judge differences, if quantities are compared.

- All the suggestions of formatting have been taken into account.

**Technical:** The authors should follow the style of the journal for example how figures are referenced, dates are given and SI units should be used.

- These points have been corrected.

- **Answers to RC3:**

1- The conclusion of this study is obviously hand waving as they conclude that the origin of mixing OH reactivity is either uncharacterized emission or oxidation products. Those are basically the nature of all VOCs in the atmosphere anyway. A deeper discussion may be utilizing a box model is recommended to narrow down the source of missing OH reactivity.

- We thank the referee for his/her suggestion. Indeed, running the model would definitely provide more insights into the origin of the missing OH reactivity, however the use of a box model is out of the scope of this paper and would require much more time (to prepare the data files, to constrain the model, to run the model and to interpret the results). Nevertheless, this idea has been added in the perspectives. Please note that section 3.5 on the investigation of missing OH reactivity was restructured, in order to make the discussion about the origins of missing OH reactivity more clear.

2- It is not entirely clear whether ambient VOC samples and OH reactivity samples were collected with the same sampling tubes. Please clarify this point as it is very important to evaluate potential imparity.

- VOC measured with the PTR-QiToFMS and used for OH reactivity calculations were sampled through 1/4"-OD PFA lines, heated at 50 ∘C and constantly flushed at 10 L min$^{-1}$ (page 12, lines 363- 364, revised version of the paper). For OH reactivity measurements, samples were collected through 1/8"-OD sulfinert lines, heated up to 50 ∘C with a sampling flow rate of 1- 1.2 L min$^{-1}$ (page 7, lines 199- 205). As mentioned in the answer to referee 2, comments 2 and 3, all lines were heated up to 50 ∘C, so no losses of VOCs are expected.

3- As the oxidation product of CO is HO2, it is more likely susceptible to interference from OH recycling during the calibration process with high CO concentrations. What CO levels do you use for calibration? Could you provide at least simple discussion that was not the case in your calibration process?

- It was initially mentioned in the text: "The measurements with CO do not correspond to a calibration procedure as the UL-FAGE instrument provides directly OH reactivity from a mono-exponential fit of the OH decay measured. It is a systematic procedure to check that the instrument provides consistent reactivity values. For that, a mixture of humid dry air with different concentrations of CO (from 4x10$^{13}$ to 3.7x10$^{14}$ cm$^{-3}$, corresponding to OH reactivity from 10 to 90 s$^{-1}$) are injected in the photolysis cell. In absence of NO, HO$_2$ is not recycled in OH and does not interfere with the OH measurement".

This is clarified in the revised version of the text, page 9, lines 276- 282: In order to check the consistency of the OH reactivity measurements, the well-known (CO + OH) reaction rate constant was measured. Different CO concentrations, from $4 \times 10^{13}$ to $3.7 \times 10^{14}$ cm$^{-3}$ in humid zero air are injected in the photolysis cell, allowing to measure reactivities ranging from 10 to 90 s$^{-1}$ and to determine (using a linear regression: R$^2$ = 0.97) a rate constant of k$_{CO + OH}$ = (2.45 ± 0.11) $\times$ 10$^{-13}$ cm$^3$ molecule$^{-1}$ s$^{-1}$, in good agreement with the reference value of 2.31 $\times$ 10$^{-13}$ cm$^3$ molecule$^{-1}$ s$^{-1}$ (Atkinson et al., 2006) at room temperature. Under these conditions (absence of NO), HO$_2$ formed by the reaction of CO+OH is not recycled in OH and does not interfere with the measurements of OH.

4- It appears that the trace gas OH reactivity such as CO, NOx, O$_3$ and SO$_2$ is not considered in the calculated OH reactivity assessments. Considering the rural location, this may not be a substantial factor, but it still requires to be included.

- NOx measurements were only performed at 6 m height. They were not included in the initial OH reactivity calculation, as indicated in page 16, point 3 (revised version), however their contribution to OH reactivity was estimated and discussed in page 28, lines 750- 753, together with O$_3$ calculated OH reactivity and CO estimated OH reactivity. No SO$_2$ measurements were performed on site but are expected to be very low.

5- Page 13 Line 12: Further quantitative discussion on the impacts from MT to the isoprene mass. What species would be susceptible for the fragment and how prevalent it can be?

- Two papers were cited in the text in which $m/z$ 69 was found as a product ion of monoterpenes fragmentation. Tani et al., 2013, reported that the relative abundance of $m/z$ 69 from myrcene fragmentation was 3.1 % for a E/N ratio of 120- 122 Td, while Kari et al., 2018 showed that $m/z$ 69 contributes between 3.8 and 4.7 % to the total corrected cps of β-myrcene, depending on E/N (range: 80- 130 Td). Other monoterpenes that can fragment at $m/z$ 69 are monoterpene alcohols linalool and cineole (Tani et al., 2013).

6- It is well known that PTR sees higher MT then the sum of speciated MT quantified by GC. Add this discussion whether that was the case during the observational period. This may give us some insight on the missing OH reactivity.

- The PTR-MS indeed measured higher MT mixing ratios than the sum of speciated MTs quantified by GC in our study. Comparisons were done at both levels. Graphs and respective discussions are presented in the supplementary material 2 "Consistency between GC and PTR-MS for monoterpene measurements".

This information is mentioned in the text: Page 15, point 1 (revised version).

7- Page 14 Line 3: Further quantitative discussion is required. It is not clear how the 4 % value has been drawn.

In order to determine the interference level of MT on isoprene measurements by PTR-MS, correlations between isoprene concentrations measured with the GC-NMHC and with the PTR-MS have been performed for different %. The agreement observed between the corrected isoprene concentrations from PTR-MS and the isoprene concentrations measured by GC was then evaluated. It was found that subtracting 4 % of the monoterpenes concentration was leading to the best agreement between the 2 instruments for isoprene. This approach assumes that the fragmentation level of monoterpenes does not change over the whole campaign.

8- Figure 2: (a) it is extremely confusing what I should look up to for the comparison. It would be better separate into figures describing in the different periods. I would recommend to present an intercomparison figure first so that readers can get a sense on the potential bias from the instrumentation.

- We thank the referee for the suggestion. It was taken into account.

Revised version, page 17: Figure 2.

[Figure]

**Figure 2.** Time series of total OH reactivity measured by UL-FAGE (dark blue) and LSCE-CRM (light blue) instruments from the 18th to 19th of July 2017, at the same location inside canopy.

(b) Also, please make it clear which MT species are consisting the total MT presented in the figure.

Revised version of the legend of Fig. 4(b): ...The lower graph (b) shows the sum of monoterpenes (MTs) and isoprene measured with the PTR-MS, in the field for the same period. Dark blue and light blue dots correspond to isoprene concentrations at 6 and 12 m height, respectively. Orange and yellow dots represent monoterpenes concentrations at 6 and 12 m height, respectively.

[Figure]

**Figure 4.** (a) Time series of total OH reactivity measured by UL-FAGE and LSCE-CRM instruments from the 13th to 18th of July 2017 (upper graph). Dark blue symbols represent the measured reactivity by UL-FAGE, green, yellow and blue symbols represent the measured reactivity by LSCE-CRM inside canopy, above canopy and inside canopy at the same location as the UL-FAGE instrument, respectively. The lower graph (b) shows the sum of monoterpenes (MTs) and isoprene measured with the PTR-MS, in the field for the same period. Dark blue and light blue dots correspond to isoprene concentrations at 6 and 12 m height, respectively. Orange and yellow dots represent monoterpenes concentrations at 6 and 12 m height, respectively.

9- Figure 3: If you take a diurnal average and adjust the intercept, then do two diurnal variations agree better? It seems CRM has 4 s$^{-1}$ offset but the text description says otherwise. Please make them consistent! In addition, even without the intercept, there are _ 20 % differences in the relationship. Please discuss the potential reasons!

- Regarding the slope, the 20 % difference is within the uncertainty of the instruments.
- Concerning the offset, the text has been clarified (Revised version):
  Page 17, line 515- page 18, line 524: When OH reactivity measurements from LSCE-CRM are plotted versus OH reactivity measurements from UL- FAGE (Fig. 3), the linear regression exhibits a slope of 1.17± 0.02, an intercept of 4.2 ± 0.4 s$^{-1}$ and a R2 of 0.87. This high intercept is statistically significant at 3σ and can partly be due to an overestimation of the UL-FAGE zero that is subtracted to the measured ambient OH reactivity. This issue is related to the quality of zero air used for zeroing the instrument. Indeed, previous comparisons have shown that using zero air of better quality (99.999%) may result in a zero of about 2 s$^{-1}$ lower (Hansen et al., 2015). An intercomparison of OH reactivity instruments made in the SAPHIR chamber (Fuchs et al., 2017) has also shown a positive bias of 1 s$^{-1}$ for the UL-FAGE instrument when high grade zero air was flushed in the chamber. A maximum overestimation of the UL-FAGE zero by 3 s$^{-1}$ is possible for this study leading to an underestimation of the ambient OH reactivity by 3 s$^{-1}$. Finally, we cannot exclude a potential offset in LSCE-CRM measurements, that could be related to a possible desorption of "sticky" compounds from the Teflon pump.

10- A more description on u* is required: how you measured them and justify the classifications.

- The turbulence was characterized using a 3D sonic anemometer (R3, Gill instruments), localized at 15 m above ground level (Kammer et al., 2018).

The information has been added in the text:

Revised version, page 13, lines 382- 384: Meteorological parameters such as temperature, relative humidity, global radiation, vertical turbulence, wind speed and wind direction were monitored using sensors already available at the ICOS measurement site. More details can be found in Kammer et al., 2018.

- Classification criteria for stable, unstable and stable/unstable nights can be found in Kammer et al., 2018, studying new particle formation episodes at the same site. In their study, the authors reported that, when NPF episodes started, u* was always lower than 0.5 m s$^{-1}$. This may be explained by the fact that nocturnal stratification led to precursor concentration increase, favoring nocturnal gas to particle conversion. Whereas, during day-time, u* was typically higher than 0.5 m s$^{-1}$. In our study, mean u* was considered and the classification was done based on graphical observations.

11- Page 20 line 12: Have you seen the described extreme weather events during the observations? If you have not, then this discussion is irrelevant.

- Indeed, this extreme weather was observed between the 18th and the 19th, July. It was clarified in the text:

Revised version, page 22, lines 624- 627: However, it is worth noting that during this night, an intense wind, rain and thunders occurred, which could have led to the observed bursts of BVOCs (Nakashima et al., 2013), leading to distinct peaks of BVOCs and total OH reactivity and thus relatively high total OH reactivity compared to other nights from the same class.

12- The stable nocturnal boundary layer could cause accumulation of long-lived oxidation products of VOCs instead of vertical mixing. Therefore, the speculation for the MT emission attributing missing OH reactivity is not conclusive. The authors need to substantiate argument.

- Yes, the reviewer is right and the accumulation of oxidation products could explain part of the missing reactivity since a higher missing OH reactivity was observed during the night of the 4th-5th, July when continental air masses imported emissions from forests and their oxidation products. The measured species showed higher levels during this stable night, a condition that could be favorable for their accumulation, as well as other unmeasured long-lived oxidation products. The information has been added in the revised version:

Page 30, line 830- 31, line 842:

Interestingly, isoprene, acetic acid and MVK+ MACR+ISOPOOH exhibited higher concentration levels during the night of the 4th- 5th, July, which was not the case for the 6th-7th, July night. Indeed, these species marked relatively high nocturnal/ inside canopy levels. When looking at air masses backward trajectories (Fig. 10), the 4th-5th night was characterized by an air mass originally coming from the ocean, which spent at least 48 hours above the continent before reaching the site. This could have led to the enrichment of the air mass with species emitted by the widely spread Landes forests and their oxidation products. Thus, the significant missing OH reactivity observed during the mentioned night is likely related to unconsidered compounds of biogenic origin characterized by a similar behavior to that of isoprene, acetic acid and MVK+MACR+ISOPOOH, which accumulated in the stable nocturnal boundary layer. In contrast, air masses spent approximately 12-18 hours above the continent during the 6th-7th of July, with more time above the ocean. Marine air masses are generally known to be clean, with relatively low levels of reactive species. Even though, the night of the 5th-6th, July shows similar air mass backward trajectories to the night of the 4th-5th, the higher turbulence during this night prevents the accumulation of reactive species (including long-lived oxidation products) due to a higher boundary layer height, lowering the reactivity and the missing OH reactivity (Fig. 10).

**List of the minor changes made since the author's response (5th October, 2019)**

- Regarding CRM and FAGE instruments data, considered in the comparison of these instruments measurements at two different locations inside canopy, we found that a short period (30 minutes) was missing from the data set. It was corrected, which changes the mean measured OH reactivity by the CRM from 19.1 to 19.2 $s^{-1}$ (section 3.3).
  - Section 3.1.2.:

  Submitted version: CRM vs. FAGE resulted in a linear regression with a slope of 1.22, an intercept of -0.69 and a correlation coefficient of 0.85.

  Revised version: CRM vs. FAGE resulted in a linear regression with a slope of 1.26, an intercept of -1.17 and a correlation coefficient of 0.87.

- Regarding the contributions of individual compounds to calculated OH reactivity (section 3.4), we made sure that the whole measurement period was considered and that the period between the 16th July at 15h and the 17th July at 12h (due to an electrical failure as mentioned in section 3.3) was excluded. Slight corrections are shown in the following table.

|  | Submitted version | Revised version |
|---|---|---|
| Inside canopy/ Day-time contribution | 65% MTs, 27% Isoprene 3% MVK+ MACR | 68% MTs, 25% Isoprene 2% MVK+ MACR |
| Inside Canopy/ Night-time contribution | 91% MTs, 5% Isoprene 2% OVOCs | 92% MTs, 4% Isoprene 1% OVOCs |
| Above canopy/ Day-time contribution | 63% MTs and 29% isoprene 2% OVOCs | 65% MTs and 27% isoprene 3% OVOCs |
| Above canopy/ Night-time contribution | 88% MTs 7% isoprene | 89% MTs 6% isoprene |

This period from the 16th (15h) to the 17th (12h) was also excluded from data presented in Fig. 8, as well as from the correlation between SQTabove/MTsabove and SQTinside/MTsinside in section 3.5 (night-time missing OH reactivity), which corrects the slope from 0.72 (with $R^2$ of 0.5) to 0.73 (with $R^2$ of 0.6).

- Averages of measured and calculated OH reactivity, summarized in table 4 (presented in the revised version based on the suggestion made by referee 1, section 3.5), were revised (in red are the values in the response to the referee and in black the values in the revised version).

| | Mean Measured OH reactivity (s$^{-1}$) | Mean missing OH reactivity with PTRQi-ToFMS (s$^{-1}$) | Missing ROH considering PTRQi-ToFMS data + other measurements (s$^{-1}$) |
|---|---|---|---|
| **Inside** | 19.0/ 19.1 | 7.3/ 7.2 | 4.3/ 4.2 |
| Day | 18.4/ 16.8 | 7.0/ 7.3 | 4.1/ 4.7 |
| Night | 21.4/ 22.0 | 9.0/ 7.1 | 5.6/ 3.6 |
| Stable cool nights | 20.5/ 20.5 | 5.7/ 5.5 | 2.1/ <LOD |
| Stable warm nights | 41.6/ 41.6 | 10.9/ 10.7 | 6.9/ 6.7 |
| Unstable cool nights | 7.9/ 7.9 | 4.5/ 4.5 | <LOD/ <LOD |
| Unstable warm nights | 13.5/ 13.5 | 6.8/ 6.8 | 3.6/ 3.6 |
| **Above** | 12.6/ 12.8 | 6.0/ 6.1 | 4.2/ 4.3 |
| Day | 10.4/ 10.7 | 5.0/ 5.1 | 3.1/ 3.3 |
| Night | 15.5/ 15.5 | 7.5/ 7.5 | 5.6/ 5.6 |
| Stable cool nights | 14.8/ 14.8 | 7.5/ 7.5 | 5.7/ 5.7 |
| Stable warm nights | ____/ ____ | ____/ ____ | ____/ ____ |
| Unstable cool nights | ____/ ____ | ____/ ____ | ____/ ____ |
| Unstable warm nights | 20.5/ 20.5 | 7.1/ 7.1 | 5.2/ 5.2 |

This correction affects also the values mentioned in the response to the referees (based on this Table), that were added to the revised version of the manuscript (page 28, section 3.5):

- Line 749: 7.2 instead of 7.3 s$^{-1}$, and 6.1 s$^{-1}$ instead of 6.0 s$^{-1}$
- Line 753: 4.2 s$^{-1}$ instead of 4.3 s$^{-1}$
- Line 763: 4.3 s$^{-1}$ instead of 4.2 s$^{-1}$
- Page 29, line 773: 7.5 s$^{-1}$ instead of 7.6 s$^{-1}$
- Table in supplementary material 9 (revised version).

In addition, for consistency only averaged contributions of NMHCs and OVOCs corresponding to the common measurements periods are presented in the revised text.

- Lines 754- 758 (revised version): 0.48 s$^{-1}$ on average, (0.43 s$^{-1}$ from NMHCs and 0.05 s$^{-1}$ from OVOCs measured by GC) of the missing OH reactivity between the 10$^{th}$ and the 12$^{th}$ July. However, after the 14$^{th}$ of July, the GC measuring OVOC stopped working, but NMHCs alone account for 0.5 s$^{-1}$ of missing OH reactivity on average.

  Instead of (page 28, lines 5- 8, submitted version): 0.45 s$^{-1}$ on average, (0.34 s$^{-1}$ from NMHCs and 0.11 s$^{-1}$ from OVOCs measured by GC) of the missing OH reactivity between the 10$^{th}$ and the 12$^{th}$ July. However, after the 14$^{th}$ of July, the GC measuring OVOC stopped working, but NMHCs alone account for 0.6 s$^{-1}$ of missing OH reactivity on average.

- Since standard OH reactivity experiments were conducted at pyrrole-to-OH ratios ranging between 1.7 and 4, for consistency, CRM data points with pyrrole-to-OH ratio > 4 were excluded from the calculation of the average increase in OH reactivity due to the different corrections (Answer to RC2, comment 1). Hence, the increase due to humidity correction is now 2.2 s$^{-1}$ instead of 2.4 s$^{-1}$, the increase due to the correction for the deviation from pseudo-first order kinetics is now 10.4 s$^{-1}$ instead of 10.0 s$^{-1}$ and the increase due to the correction for dilution is now 2.6 s$^{-1}$ instead of 2.8 s$^{-1}$ (Table 2, revised version).

- The contribution of acetic acid to OH reactivity was corrected:
  Submitted version, page 29, line 2: Maximum OH reactivity was on average 0.8 s$^{-1}$ for warm days, 3.8 times higher than for cool days (inside canopy measurements).
  Revised version, page 30, line 811: Maximum OH reactivity was on average 0.07 s$^{-1}$ for warm days, 4 times higher than for cool days (inside canopy measurements).

[revised manuscript text omitted]

---

## Author Response (AR2)

**Author's response**

Editor's report:

Comments to the Author:
Thank you for your detailed responses to the comments of the three referees. There is just one matter raised by RC2 that I would like you to address:

In the response to RC2, Point 6:
6- Figure 3: In a correlation plot, error bars of measurements are needed. Did the regression procedure take into account errors of the measurements?

Response:
Errors bars were added as shown in Fig. 3, page 18 of the revised version.

Thank you for doing this.

Errors of the measurements were not taken into account in the regression procedure.

**Can you please do this. It should be straightforward to calculate the gradient which takes into account errors on both X and Y data, for example using the orthogonal distance regression technique, or perhaps others.**

Many thanks.

*We thank the editor for evaluating the paper, and for the suggestion made on the technique that can be used to include measurements uncertainties in the correlation between LSCE-CRM and UL-FAGE data (same place, section 3.1.1).*

*The Orthogonal Distance regression technique was applied using a script run on Python and considering an uncertainty of 35% and 15% for the CRM and the FAGE instrument, respectively. The following values for the slope and the intercept are obtained:*

- *Revised version: slope = 1.28 ± 0.02, intercept= 0.96 ± 0.23*
- *Older version: slope = 1.17 ± 0.02, intercept= 4.22 ± 0.41 with a R2 of 0.87*

*Thus, the resulting slope is still within measurement uncertainties and the intercept is not significant anymore.*

*Since the slope and the intercept of the correlation between LSCE-CRM and UL-FAGE instruments at two different places inside the canopy are also presented (section 3.1.2), we found that it was more consistent to use the Orthogonal distance regression technique also on these measurements. We obtained the following:*

*- Revised version: slope= 1.28 ± 0.02, intercept= -2.63 ± 0.15*

*- Older version: slope= 1.26 ± 0.01, intercept = -1.17 ± 0.17 with a R2 of 0.87*

*Changes were made in the text in sections 3.1.1, 3.1.2, 3.2, 3.5 and 4. They are presented in the following sections. We note that this change does not induce any change in the other results and their discussion.*

[revised manuscript text omitted]